# Holocene vegetation dynamics in response to climate change and hydrological processes in the Bohai region

Chen Jinxia[a,b,*], Shi Xuefa[a,b,*], Liu Yanguang[a,b], Qiao Shuqing[a,b], Yang Shixiong[c,d], Yan Shijuan[a,b], Lv Huahua[a,b], Li Jianyong[e,f], Li Xiaoyan[a,b], Li Chaoxin[a,b]

[a] Key Laboratory of Marine Geology and Metallogeny, First Institute of Oceanography, MNR, Qingdao 266061, China

[b] Laboratory for Marine Geology, Pilot National Laboratory for Marine Science and Technology, Qingdao 266061, China

[c] Key Laboratory of Coastal Wetland Biogeosciences, China Geological Survey, Qingdao 266071, Shandong, China

[d] Laboratory for Marine Geology, Qingdao National Laboratory for Marine Science and Technology, Qingdao, 266061, China

[e] Shanxi Key Laboratory of Earth Surface System and Environmental Carrying Capacity, College of Urban and Environmental Sciences, Northwest University, Xi'an, China

[f] Institute of Earth Surface System and Hazards, College of Urban and Environmental Sciences, Northwest University, Xi'an, China,

[*] Corresponding author at: Key Laboratory of Marine Geology and Metallogeny, First Institute of Oceanography, MNR, Qingdao 266061, China.

*E-mail address*: jinxiachen@fio.org.cn (J. Chen); xfshi@fio.org.cn (X. Shi).

## ABSTRACT

Coastal vegetation both mitigates the damage inflicted by marine disasters on coastal areas and plays an important role in the global carbon cycle (i.e. blue carbon). Nevertheless, detailed records of changes in coastal vegetation composition and diversity in the Holocene, coupled with climate change and river evolution, remain unclear. To explore vegetation dynamics and their influencing factors on the coastal area of the Bohai Sea (BS) during the Holocene, we present high-resolution pollen and sediment grain size data obtained from a sediment core of the BS. The results reveal that two rapid and abrupt changes in salt marsh vegetation are linked with the river-system changes. Within each event, a recurring pattern—starting with a decline in Cyperaceae, followed by an increase in *Artemisia* and Chenopodiaceae—suggests a successional process that is determined by the close relationship between Yellow River (YR) channel shifts and the wetland community dynamics. The phreatophyte Cyperaceae at the base of each sequence indicate lower saline conditions. Unchannelized river flow characterized the onset of the YR channel shift, caused a huge river-derived sediment accumulation in the floodplain, and destroyed the sedges in the coastal depression. Along with the formation of a new channel, lateral migration of the lower channel stopped, and a new intertidal mudflat was formed. Pioneer species (Chenopodiaceae, *Artemisia*) were the first to colonize the bare zones of the lower and middle marsh areas. In addition, the pollen

results revealed that the vegetation of the BS land area was dominated by broadleaved forests during
the early Holocene (8500–6500 a BP) and by conifer and broadleaved forests in the middle Holocene
(6500–3500 a BP), which was followed by an expansion of broadleaved trees (after 3500 a BP). The
pollen record indicated that a warmer early and late Holocene and colder middle Holocene were
consistent with previously reported temperature records for East Asia. The main driving factors of
temperature variation in this region are insolation, the El Niño-Southern Oscillation and greenhouse
gases forcing.
*Keywords*: Coastal salt marsh; Pollen; Delta superlobe; Temperature; El Niño-Southern Oscillation

## 1. Introduction

Coastal areas, where cities, populations and industries are clustered, are playing an increasingly

critical role in trade globalization (Hemavathi et al., 2019). Because they are located between marine
ecosystems and terrestrial ecosystems, coastal areas are prone to many natural hazards such as
flooding, storms, and tsunamis (Hou and Hou, 2020). Coastal vegetation, which acts as a natural
barrier, is widely distributed in coastal areas and could effectively mitigate the damage caused by
marine disasters to the economy and environment of coastal areas (Zhang et al., 2018). Moreover,
despite their relatively small global extent (between 0.5 and $1 \times 10^6$ km$^2$), coastal vegetation
ecosystems, tidal marshes, mangroves, and seagrasses play an important role in the global carbon
cycle (Serrano et al., 2019; Spivak et al., 2019). Per unit area, their organic carbon sequestration
rates exceed those of terrestrial forests by 1–2 orders of magnitude and contribute ~50% of carbon
sequestered in marine sediments (Serrano et al., 2019). Hence, it is important to understand the long-
term spatial–temporal dynamics of coastal vegetation, which are favorable for the global carbon
cycle research and coastal restoration.

Climatic fluctuation, post glacial sea-level rise and changes in river discharge provoked

dramatic habitat changes along coastal areas during the Late Pleistocene and Holocene (Neumann
et al., 2010; Cohen et al., 2012; Pessenda et al., 2012; França et al., 2015). Presently, the relationship
of sea-level change and coastal vegetation (especially mangrove) evolution has been widely studied
by many researchers (e.g. Engelhart et al., 2007; Gonzalez and Dupont, 2009; França et al., 2012;
Woodroffe et al., 2015; Hendy et al., 2016). Contrarily, studies on the long-term dynamics of coastal

vegetation, coupled with climate change and river evolution, are sparse. During the Holocene, the global rivers delivered large amounts of material to the ocean, the total suspended sediment delivered by all rivers to the ocean was approximately $13.5 \times 10^9$ tons annually (Milliman and Meade, 1983). The material transported by the rivers had huge impacts on the coastal ecosystem. Hence, a deeper understanding of correlations between coastal vegetation and river variables is required to better assess coastal vegetation responses to global warming in the future.

In the coastal areas of the Bohai Sea (BS), vegetation is dominated by warm temperate deciduous broadleaved forests and shrub grasslands (Wang et al., 1993). The Yellow River (YR), as one of the largest river in the world in terms of sediment discharge (Milliman and Meade, 1983), transports large amounts of sediment into the BS every year; hence, it has developed a delta complex in the west coastal region of the BS since 7000 a BP (He et al., 2019). Deposition of the Yellow River delta (YRD) complex resulted in the formation of a vast area of floodplain and estuarine wetland (Xue et al., 1995; Cui et al., 2009; Liu et al., 2009b). Based on the study of coastal vegetation of the BS, it is helpful to understand the spatial and temporal drivers of ecological variability, and thus of the vegetation-climate and river relationship, especially wetland dynamics. However, there have been few studies investigating the vegetation dynamics and their response to climate and river variables in the Bohai region.

Pollen records have been useful in terms of reconstructing vegetation dynamics and environmental changes associated with climatic changes in the geological record (Bao et al., 2007; Cohen et al., 2008; Giraldo-Giraldo et al., 2018). In this study, we carried out a detailed investigation of core sediments from Laizhou Bay, BS. We analyzed pollen and grain size proxies under high resolution and refined the chronology of the core by $^{137}$Cs and accelerator mass spectrometry (AMS) $^{14}$C dates. With this in mind, the specific objectives of the current research are formulated as follows: (1) to reconstruct the vegetation evolution history in the Bohai region and (2) to tentatively discuss the effects of climate and environment on coastal vegetation (especially wetlands) during the Holocene.

## 2. Study area

### 2.1. Geographical settings

The BS, a shallow inland sea in China, is connected with the Yellow Sea through the narrow

Bohai Strait (Figure 1). The main rivers flowing into the BS are the YR, Haihe River, Luanhe River,
and Liaohe River. Among these, the YR is the largest and is the main source of sediments in this
region. Over the past 2000 years, the YR has annually provided approximately $1.1 \times 10^9$ tons of
sediment discharged into the BS (Milliman et al., 1987). This immense amount of sediment has
resulted in the rapid seaward progradation of YRD, and a rapid change in the location of the main
distributaries in the lower delta plain.

The tidal current plays a critical role in the transportation and distribution of sediments in the

BS. The tidal currents of the modern BS are dominated by semi-diurnal tides. The velocity of tidal
currents varies from 20 to 80 cm/s. Three strong tidal current areas are observed in the northern
Bohai Strait, the central part of Bohai Bay, and the eastern part of Liaodong Bay (Huang et al., 1999).
In Laizhou Bay close to the core location, the speed of tidal currents is weak (Gu and Xiu, 1996).

The wind waves off the YRD are dominated by the East Asian monsoon and show significant

seasonal variations. The prevailing northerly winds are much stronger in winter than the dominant
southerly winds in summer. Strong winter winds cause strong wind waves, and thus strong bottom
shear stresses which readily erode seabed sediment into water (Yang et al., 2011; Wang et al., 2014;
Zhou et al., 2017).

The circulation of the BS is weak, and the mean flow velocity is small. In winter, the

predominant extension of the Yellow Sea Warm Current (YSWC) intrudes and crosses the Bohai
Strait, moving westward along the central part of the BS and splits into two branches. One branch
moves toward the northeast to form a clockwise gyre (Liaoxi Coastal Current [LXCC]), and the
other veers southward and then turns eastward along the southern coast to form a counterclockwise
gyre (Libei Coastal Current [LBCC]). In summer, the YSWC disappears in the BS, and eddies
generated in the BS are stronger than in winter. During this time, the central eddy is missing, the
eddy in Laizhou Bay is more pronounced, and the coastal current along the southern and western
coastlines of the BS is established (Figure 2; Liu et al., 2015; Yang et al., 2016).

**2.2. Climate and vegetation**

The Bohai region lies in a zone of warm temperate monsoonal climate with distinctive seasons.

The annual mean air temperature is 9.5–13.1 °C. The annual average precipitation is about 600 mm,
and 60–70% of the total annual precipitation occurs between June and August (Qiao et al., 2012).
As the Liaodong Peninsula and Shandong Peninsula protrude into the sea, they are clearly
influenced by its proximity and experience sufficient rainfall. There is less rainfall in the mountain
area of the northern part (Wang et al., 1993).
The regional vegetation is dominated by warm temperature deciduous broadleaved forests and
shrub grasslands. Currently, natural vegetation only remains in the mountain areas because of
widespread anthropogenic activities (e.g. cultivation and farming). The predominant deciduous
broadleaved species belong to *Quercus*, such as *Q. liaotungensis*, *Q. dentata, Q. acutissima*, and *Q.*
*mongolica*. Co-dominant plants are *Pinus*, including *P. densiflora* that grows in the coastal humid
area, and *P. tabuliformis* that is distributed in the relatively dry North China plain. In the plain area,
apart from *P. tabuliformis*, there are some deciduous broadleaved trees, such as *Ailanthus altissima*,
*Koelreuteria paniculata*, and *Morus alba*. Other broadleaved trees, *Betula ermanii*, *Populus tremula*,
*Acer* spp., *Tilia amurensis*, and *Carpinus turczaninowii* are distributed in the hills and lowlands
(Wang et al., 1993). The coastal wetlands are occupied by herbs and shrubs, such as *Tamarix*
*chinensis*, *Salix matsudana*, *S. integra*, *Phragmites australis*, *Aeluropus*, *Limonium sinense*, *Suaeda*
*glauca*, *Typha orientalis*, and *Acorus calamus* (Wang et al., 1993; Li et al., 2007; Xu et al., 2010).

**3. Materials and methods**
**3.1. Coring, sub-sampling, and chronology**
Core CJ06-435 was collected in Laizhou Bay, BS in August 2007 by the R/V *Kan407* of the
Shanghai Bureau. The core site is located at 37.50 °N, 119.52 °E, at a water depth of 14.6 m (Figure
1); the core had a length of 271 cm. In the laboratory, the core was spilt into two sections,
photographed, macroscopically described, and sub-sampled.
Isotopes [137]Cs and [210]Pb were measured employing EG&G Ortec Gamma Spectrometry at the
Nanjing Institute of Geography and Limnology, Chinese Academy of Sciences (NIGLAS). The
sediment samples were air-dried and pulverized. [137]Cs and [210]Pb concentrations were then
determined from gamma emissions at 662 and 46.5 keV, respectively. In addition, a total of 10
samples consisting of foraminifera were obtained from the core for radiocarbon dating. The
radiocarbon dating was conducted at the Woods Hole Oceanographic Institution (WHOI) and Beta
Analytic Inc., USA. Radiocarbon dates were corrected for the regional marine reservoir effect (ΔR
= −139 ±59 years, a regional average value determined for the BS) and calibrated using the Calib
7.04 program (Stuiver et al., 2019) with one standard deviation uncertainty ($1.0 \times \sigma$) (Table 1).

**3.2. Palynological and sediment grain size sample analysis**
A total of 127 samples were selected for pollen analyses. Each sample was oven-dried at 60 ℃
for 24 h. The dry-weight of the samples ranged from 2.5 g to 13.9 g. Samples were chemically
treated according to the procedure outlined by Faegri and Iversen (1992). Before treatment, a
standard tablet of *Lycopodium* spores (mean=18,583$\pm$764 spores per tablet) was added to each
sample to aid in the calculation of palynological concentrations. Samples were treated with 15%
HCl solution to remove carbonates, boiled in 10% KOH solution for 5 min to remove humic acids,
then treated with 40% HF to remove silicates. The residue was mounted in glycerin jelly. Fossil
pollen was identified and counted with a light microscope at 400$\times$ magnification. A minimum of
200 pollen grains were counted for each sample. The palynological concentrations of per gram
sediment (PCP) were calculated using the follow equation:
$$PCP = \frac{18583}{Lycopodium \text{ number per slide}} * \frac{\text{Pollen or Spore Counts per slide}}{\text{Net weight of dry sample}}$$
The percentage of each pollen type was calculated from the total sum of pollen and spores.
The pollen diagram was produced using the Tilia program, and the pollen assemblage zones were
divided based on the results of a constrained cluster analysis (CONISS) within Tilia (Grimm, 1987).
Analysis of sediment grain size was performed at 2.0 cm intervals throughout the core using
a Malvern Mastersizer 2000 instrument at the laboratory of the First Institute of Oceanography. The
chemical procedure of grain size experimental pretreatment was consistent with the procedures
described by Chen et al. (2019a). A solution of 30% $H_2O_2$ and 1.0 mol/L HCl were added to
decompose the organic matter and remove carbonates.

**4. Results**
**4.1. Chronological model**
Measurements of [137]Cs and [210]Pb revealed activity at the top of the profile, indicating the
recovery of recently deposited sediments. [137]Cs is a bomb-derived radionuclide, first appearing in
environmental samples at measurable levels around 1954 with the onset of nuclear weapon testing
(Kirchner and Ehlers, 1998), and was most prevalent in 1963 (the year of maximum fallout from
atmospheric weapon testing) (Palinkas and Nittrouer, 2007). Sub-surface peaks are not discernible
in [137]Cs profiles of core CJ06-435 (Figure 3). However, the deepest onset of [137]Cs is an effective
marker of the year 1954 (25 cm).
Often, the combined data of [137]Cs and excess [210]Pb have been used to calculate the
sedimentation rates (Wu et al., 2015). Excess [210]Pb shows a downward decline owing to the decay
of [210]Pb when the sediment stably accumulates for an appropriate period, and the excess [210]Pb
activity could be used to calculate the sedimentation rate. However, the excess [210]Pb profiles of core
CJ06-435 did not show a clear downward decline trend (Figure 3), and excess [210]Pb in the upper
parts of the core is not that large when compared with the lower background values. Thus, the [210]Pb
data seemed to be unsuitable for estimating the sedimentation rate of core CJ06-435. The [137]Cs–
derived average sedimentation rate was 0.47 cm/yr in the upper 25 cm of core CJ06-435.
The results of AMS radiocarbon dating are shown in Table 1 and Figure 3. Three samples
above the 20 cm depth were not included in the age model because their [14]C age was anomalously
greater than the [137]Cs dating. The dating point of 129 cm was eliminated because it appears not to
be reliable. According to the result of He et al. (2019), the calculated sedimentation rate (CSR) in
the tidal flat and neritic area of the south BS ranged from 0.02 to 0.13 cm/year before 2000 a BP
(calculation from cores H9601, H9602, ZK228, and ZK1, Figure 1). If the 129 cm dating is correct,
the CSR would be as high as 0.45 cm/yr in the section of 87–129 cm (4801–4894 a BP) for core
CJ06-435. It is apparently not reasonable because core CJ06-435 is offshore compared to the other
cores (e.g. H9601, H9602, ZK228, and ZK1, Figure 1) reported in previous researches (Xue et al.,
1988; Saito et al., 2000; Li et al., 2013). It should have a lower CSR compared to those cores rather
than an approximate ten-fold increase in CSR. The calibrated dates of several other samples are
plotted against sediment depth and shown in Figure 4.

**4.2. Sediment grain size distributions**
The grain size parameters and components percentages show distinct variations. The mean
grain size and the median grain size both show high values at depths of 271–160 cm, 135–83 cm
and 19–0 cm, lower values at a depth of 83–34 cm, and the lowest values at depths of 160–135 cm
and 34–19 cm. There was a smaller proportion of clay in the lower profile (271–160 cm) than in the
upper profile (160–0 cm, except for the two sections of 160–135 cm and 34–19 cm). The sequences
of silt and sand contents showed a strong inverse association. There were high proportions of silt
and low proportions of sand at depths of 271–222 cm, 180–160 cm, 135–83 cm, 40–34 cm and 19–
0 cm, lower proportions of silt and higher proportions of sand at depths of 222–180 cm and 83–40
cm, and the lowest proportions of silt and the highest proportions of sand occurred at depths of 160–
135 cm and 34–19 cm (Figure 3).

**4.3. Palynology assemblage**
A total of 71 pollen taxa were identified, among which *Pinus, Quercus*, Cyperaceae and *Typha*
were the most dominant taxa in the lower part (271–156 cm) of the core, and *Pinus*, *Quercus*,
Poaceae, Compositae, *Artemisia*, Chenopodiaceae and Cyperaceae were the most dominant taxa in
the upper part (156–0 cm) of the core. With respect to the fern spores, *Selaginella sinensis* and
Polypodiaceae were dominant; however, their content was low throughout the core. With the aid of
CONISS, the whole sequence was vertically divided into three zones, with zone 2 further
divided into subzones 2a, 2b, 2c and 2d (Figures 4 and 5).

**4.3.1. Palynological zone 1 (271–156 cm)**
The palynological zone 1 was characterized by abundant broadleaved trees pollen, dominated
by *Quercus* (mean 18.7%), *Betula*, *Alnus*, *Pterocarya*, Ulmaceae and Moraceae (Figure 4).
Percentages of conifer pollen were relatively low compared with other zones: *Pinus* ranges from
19.7% to 45.6% (mean 33.6%), and Taxodiaceae was present only occasionally. Compared to the
other zones, the proportions of non-arboreal pollen types, Compositae (mean 1.2%), *Artemisia*
(mean 4.2%), and Chenopodiaceae (mean 5.6%) were lowest in this zone, whereas the proportions
of Cyperaceae (mean 10.3%) and *Typha* (mean 11.2%) were highest in this zone. The palynological
concentrations were high, varying between 6050 and 237 grains/g (Figure 5).

**4.3.2. Palynological zone 2 (156–30 cm)**
Palynological zone 2 was divided into four subzones:
From depth of 156 to 128 cm (subzone 2a), the percentage of *Pinus* pollen reached its
maximum (mean 46.6%), whereas the percentage of broadleaved trees *Quercus* (16.1–9.5%, mean
14%), *Betula*, *Alnus*, *Pterocarya*, Ulmaceae, and Moraceae decreased to different degrees. The
proportions of non-arboreal pollen types, Compositae (mean 2.5%), *Artemisia* (mean 6.6%), and
Chenopodiaceae (mean 7.4%) were higher, whereas the percentages of Cyperaceae (mean 7.2%)
and *Typha* (mean 3.1%) declined sharply (Figure 4). Total pollen concentration were lower than in
zone 1, especially in the interval of 156–135 cm, where the value of total pollen concentrations (62–
1306 grains/g, mean 485 grains/g) was at its minimum in the core (Figure 5).
From depth of 128 to 63 cm (subzone 2b), *Pinus* (49.4–27.3%) and *Quercus* (18.1–7.9%, mean
13.5%) pollen level decreased to a low point, and the relative abundance of *Betula* slightly increased.
Occasionally, there were small amounts of *Pterocarya*, Ulmaceae, and Moraceae. Non-arboreal
pollen types, Compositae, *Artemisia*, and Chenopodiaceae continuously increased to averages of
3.5%, 6.7%, and 12%, respectively (Figure 4). Pollen concentration increased up to a high
abundance (mean 1260 grains/g) in this subzone (Figure 5).
From depth of 63 to 41 cm (subzone 2c), the percentage of *Pinus* pollen started to decrease
steadily, and the percentage of *Quercus* (17.4–11.8%, mean 14.8%), *Betula*, *Alnus*, *Pterocarya*, and
Ulmaceae pollen slightly increased. Similar to subzone 2b, this subzone had relatively high
quantities of non-arboreal pollen such as Compositae, *Artemisia*, and Chenopodiaceae (Figure 4).
Pollen concentrations varied between 456 and 1381 grains/g (Figure 5).
In contrast, subzone 2d (41–30 cm) was marked by a sudden decrease in the pollen of *Quercus*
(9.8%), and a steep increase in the pollen of *Pinus* (41.1%), even though the percentage of the non-
arboreal were the same as those of subzone 2c (Figure 4).

**4.3.3. Palynological zone 3 (30–0 cm)**
This zone was characterized by the transition from dominance by the pollen of arboreal taxa
to non-arboreal types. The percentage of *Pinus* and *Quercus* pollen decreased to the lowest level,
averaging approximately 19.7% and 5.5%, respectively. Poaceae, Compositae, *Artemisia*, and
Chenopodiaceae pollen increased, with average values of *Artemisia* and Chenopodiaceae reaching
up to 24.6% and 21.1%, respectively (Figure 4). The total pollen concentration declined to 188
grains/g in the lower part of the zone (30–19 cm) and then increased slightly (to approximately 621
grains/g) at the top (Figure 5).

**5. Discussion**

**5.1. Key terrestrial palynomorphs proxies of environmental and climatic change**

In sediment core CJ06–435, both *Pinus* and *Quercus* pollen were the predominant pollen types among the arboreal taxa. In order to understand the pollen provenance, the pollen records in the surface sediments of Laizhou Bay were studied (Yang et al., 2016), and the concentration and percentage data of main pollen species were presented on a regional map (Figure 6). Pollen results of surface sediments revealed that higher values of *Pinus* and *Quercus* are usually found in the eastern part of Laizhou Bay, and the lowest values of *Pinus* and *Quercus* occur in the nearshore area outside the mouth of the YR (Figure 6a and 6b). The distributions of *Pinus* and *Quercus* pollen in the surface sediments of Laizhou Bay are closely related to the distribution of the nearshore epicontinental vegetation. Except for the YRD, where there is swamp and cultivated land, the epicontinental region of the Laizhou Bay is surrounded by pine and oak forests. Among these, the land to the east of the Laizhou Bay (the Shandong Peninsula) belongs to the southern warm temperate zone and mainly supports a pine–oak forest dominated by *Pinus densiflora* and *Q. acutissima* (Wang et al., 1993). The land to the northeast of the Laizhou Bay (the Liaodong Peninsula) belongs to the southern temperate zone, and it mainly developing a conifer and broadleaved mixed forest dominated by *P. densiflora*, *Q. mongolica* and *Q. acutissima* (Li et al., 2007; Xu et al., 2010). The ubiquitous distribution of these plants on the adjacent terrain explains why *Pinus* and *Quercus* are the most common pollen taxa in Laizhou Bay, and why the highest concentration and percentage of *Pinus* and *Quercus* occurred in the eastern part of Laizhou Bay and the lowest values of *Pinus* and *Quercus* occurred on the nearshore area outside the mouth of the YR.

Previous studies have revealed that *Pinus* and *Quercus* were the most common components of the forests in northeast China (including the land areas surrounding the BS) during the Holocene. The variation of *Pinus* and *Quercus* contents were closely related to the change of temperature (Ren and Zhang, 1998; Yi et al., 2003; Li et al., 2004; Xu et al., 2014; Zhang et al., 2019). Ren and Zhang (1998) investigated pollen data from northeast China and found that *Quercus* and *Ulmus* were the dominant components of the forests between 10000 and 5000 a BP, while *Pinus* were much sparser, indicating warmer and drier summers in northeast China for the early to mid-Holocene. A high-resolution 1000-year pollen record from the Sanjiaowan Marr Lake in northeast China revealed that *Quercus* is an effective indicator for temperature reconstructions. Several notable cold periods, with

lower *Quercus* frequencies, occurred at approximately 1200 AD, 1410 AD, 1580 AD, 1770 AD and
1870 AD (Zhang et al., 2019). Another 5350-year pollen record from an annually laminated maar
lake in northeast China revealed a decrease of *Quercus* and increases of the *Pinus* component; this
indicates a cooling trend during the past 5350 years (Xu et al., 2014). Based on these results, we
conclude that the variation of *Pinus* and *Quercus* pollen of core CJ06-435 may be also related to
temperature change.

Herb pollen, especially Chenopodiaceae, also occupies an important position in core CJ06-435

(Figure 4). The spatial distribution of herb pollen in surface sediment of Laizhou Bay suggests that
a higher percentage and concentration occur in the nearshore area close to the YR estuary and the
southwestern part of Laizhou Bay, and a low percentage and concentration are found in the eastern
part of Laizhou Bay (Figure 6c and 6d). The YR is the main sediment source of the BS. The annual
mean sediment load of the YR was $1.08 \times 10^9$ tons before dam construction (Milliman and Meade,
1983), 70–90% of which was deposited and formed a huge delta complex (Zhou et al., 2016).
Natural vegetation in the modern YRD are dominated by wetland herbs, including Chenopodiaceae
and *Artemisia* (Jiang et al., 2013). Furthermore, under the combined action of the ocean and rivers,
alluvial plains and coast plains developed widely along the southern coast of Laizhou Bay. The
terrestrial vegetation types in these areas change from a bare intertidal zone to seepweed swamp to
reed swamp to cultivated land from the shoreline landward (Xu et al., 2010). Since the transportation
distance for herb pollen is normally very short, the pollen percentage and concentration in samples
close to the mouth of the YR and the southwestern part of Laizhou Bay are much higher than in
other samples (Figure 6c and 6d), indicating that herb pollen of Laizhou Bay is mainly derived from
the plant communities of the coastal wetlands.

It is worth noting that the composition of fossil pollen in sediment depends not only on the

composition of the vegetation from which the pollen originates but also on pollen dispersion,
deposition and preservation. *Pinus* pollen is a bisaccate grain, and has relatively high aerodynamic
and hydrodynamic characteristics meaning it can be transported efficiently by wind and water
(Sander, 2001; Montade et al., 2011). Previous studies revealed that smaller amount of *Pinus* pollen
are found nearshore, and larger amounts are found in the deep ocean (Mudie, 1982; Mudie and
McCarthy, 1994; Zheng et al., 2011; Dai et al., 2014; Luo et al., 2014; Dai and Weng, 2015). In
Laizhou Bay, although the concentration and percentage of *Pinus* pollen followed similar patterns
of distribution to broadleaved tree pollen (*Quercus*, *Betula* and *Carpinus*; Yang et al. 2016).
However, in the eastern part of Laizhou Bay, *Pinus* pollen increased in a northeasterly direction
away from the coast (Figure 6a). Hence, concerning *Pinus* pollen data, caution is required because
climate variation alone may not be responsible for the change of *Pinus* pollen in marine sediment.
Aerodynamic and hydrodynamic conditions may also influence the amounnt of *Pinus* pollen in
sediments.

In addition, because different pollen types are not equally well preserved (Havinga, 1967;

Cheddadi and Rossignol-Strick, 1995), bias originating from poor preservation should be eliminated
before using the net content of pollen grains to reconstruct paleovegetation. In this study, the pollen
concentration ranged from 62 to 6050 grains/g. Relatively low pollen concentrations were found in
the two sections (160–135 cm and 34–19 cm); this was largely correlated to high sand contents as
revealed by the lithology. Especially for the lower section (150–135 cm), the high portion of sand
content is consistent with a low pollen concentration and a high percentage of *Pinus* pollen. As the
*Pinus* pollen is more resistant to degradation, the variations of total pollen concentration as well as
a higher percentage of *Pinus* pollen in this section seem to be related to pollen preservation. But, as
shown in Figure 4, the highest percentage of *Pinus* pollen was recorded at a depth of 150–128 cm,
with a low value at 148 cm, which is not completely in accordance with the high sand content section
in the same core (160–135 cm). Similarly, for the upper section, a high sand content was recorded
at a depth of 34–19 cm. However, the percentage of *Pinus* pollen is low in this section, except for a
relatively high value at a depth of 23 cm. We thus suggest degradation is not a key point influencing
the concentration of pollen and spore in the study area.

Previous research suggests that the sedimentation mechanisms of pollen and spore in marine

water is similar to that of sediment with clay and fine silt grain size (Heusser, 1988). A recent
investigation on the surface sediment from the BS shows high pollen concentration in sediments
with a high proportion of fine particles such as clay and silty clay, while low pollen concentration
in sediments with a high proportion of coarser sand particles (Yang et al., 2019). Yang et al. (2019)
attributed the low pollen concentration in areas with a high sand content of the BS to the strong
hydrodynamic suspension and screening for sediments and pollen. We conclude that the low pollen
concentrations in the two sections (160–135 cm, and 34–19 cm), correlated with high sand content,
could be attributed to the hydrodynamic conditions rather than degradation.

**5.2. Sedimentary records indicative of river channel shifts**

The most important geological events in the northern China coast after 7000 a BP were the shift of YR channel and the formation of the YRD. The YR has been easily plugged and breached, and therefore its lower reaches migrated because of its huge sediment load. The shifting of the lower reaches of the YR led to the formation of a new delta superlobe (He et al., 2019). Based on a study of cheniers and historical documents, nine YRD superlobes have been proposed by Xue and Cheng (1989) and Xue (1993) on the western shore of the BS. Among these, superlobe 1 (7000–5000 a BP, He et al., 2019), superlobe 7 (11 AD–1048 AD, Xue, 1993), and superlobe 10 (1855 AD–present) are positioned near the core area in this study. The information about some of these superlobes formation are recorded in core CJ06-435.

As shown in Figure 8, herb percentage sudden change at 160 cm and 34 cm. Herb pollen in the sediment of Laizhou Bay is mainly derived from the coastal wetlands of the western BS. In 6000–7000 a BP and 1855 AD, the YR emptied into the BS after a natural course shift, forming two huge delta superlobes in the western part of the BS (Saito et al., 2000). Wetland plants are the most important vegetation type in the YRD (Jiang et al., 2013). The development of YRD wetland would change the amounts of herb pollen that was transported to the study site. In addition, the formation of YRD caused the coastline to move closer to the position of the CJ06-435 core. Since most herb plants are small in size, their pollen grains are unable to disperse broadly (Chen et al., 2019b). The migration of the coastline would change the availability of herb pollen to the study site, and hence lead to variations in the amount of pollen. Therefore, combined with the age data, we conclude that the abrupt change of herb pollen percentage at 160 cm and 34 cm in core CJ06-435 is related to the formation of the YRD superlobe 1 and superlobe 10.

Compared with the pollen percentage, the pollen concentration can be interpreted in different ways. Namely, the percentage of different types of pollen is relative, whereas the pollen concentration is absolute, and it can directly reflect the amounts of pollen that were transported to the study area (Luo et al., 2013). It is crucial that a correct interpretation of pollen data is based on a percentage diagram as well as concentration. In core CJ06-435, the concentrations of herbs— especially Chenopodiaceae and *Artemisia* (Figure 7f)—were higher at depths of 160–94 cm (6570– 5000 a BP) and 34–0 cm (after 1855 AD), except for the two sections of 160–135 cm and 34–19

cm. As mentioned in section 5.1, the extremely low pollen concentration in the sections of 160–135
cm and 34–19 cm was closely linked with the coarser sandy sediment. Combined with the results
of pollen percentage and sediment grain size, we presumed that the higher herb pollen concentration
in the periods of 6570–5000 a BP (160–94 cm) and after 1855 AD (34–0 cm) reflects changes in
hydrographic conditions. Pollen data of surface sediments revealed that higher herb pollen
concentrations occur in the YR, and the value of these concentrations showed a decreasing trend
starting from the river mouth toward the ocean. The distribution pattern of surface pollen revealed
that the YR is a major carrier for most herbs taxa in the sediment of Laizhou Bay (Yang et al., 2016).
At the site of core CJ06-435, which is close to the mouth of the YR in Laizhou Bay, higher herb
pollen concentrations in the Holocene samples may indicate increased fluvial discharge.
Sediment grain size provides direct information on changes of the sediment source and the
sedimentary environment (Friedman and Sanders, 1978; Wu et al., 2015). The characteristics of
grain size can be expressed by the grain size distribution curve, and usually the mean or median
diameter is used (Xu, 1999). In this study, the value of mean grain size (Mz) showed that two major
grain size boundaries occur at depths of 34 and 19 cm, separating a middle sedimentary unit (34–19
cm) that contains coarser sediment from the lower and upper sedimentary units that contain finer
sediment (Figure 7d). The sand content of the upper, middle and lower layers was 11.2%, 33.6%,
and 9%, respectively (Figure 7a); the silt content of these layers was 69%, 58.6%, and 76.1%,
respectively (Figure 7b), and the clay content of these layers was 19.8%, 7.8%, and 14.8%,
respectively (Figure 7c). On the basis of $^{137}$Cs chronology (Figure 3), we speculate that these
significant changes of grain size parameters at depths of 34 and 19 cm might represent a record of
the channel shifts of the YR in 1855 AD and 1976 AD, respectively.
The sediment of Laizhou Bay mainly comes from the YR and other small rivers located in
the southern part of Laizhou Bay (Zhang et al., 2017; Gao et al., 2018). Prior to 1855 AD, when the
YR entered the Yellow Sea, the sediment contribution to the BS from other small rivers was
relatively larger. The fine fraction suspension sediment that was derived from other small rivers
favors the hypothesis of fine sediment accumulation in core CJ06-435 during this period.
When the YR reentered the BS after 1855 AD, the dispersal of YR material contributed
substantially to the sedimentation of the BS. It was reported that more than 80% of the YR sediment
discharges into the BS during the summer period (Bi et al., 2011). Owing to the barrier effect of the
tidal shear front and the weak river flow, most of the river-delivered sediment is deposited on the
offshore delta within 15 km of the river mouth (Wang et al., 2007; Bi et al., 2010). Only a small part
of the fine clay fraction is transported by the coastal currents over long distances and deposited
across or along the shore in summer (Wu et al., 2015). During the winter (October to March) season,
the much stronger winter monsoon generates large waves, resulting in intensive sediment
resuspension in the coastal region owing to the enhanced bottom shear stress (Yang et al., 2011; Bi
et al., 2011). The resuspended sediment is transported southeastward along the coast of Laizhou Bay
by the monsoon-enhanced coastal currents passing through the location of the sediment core CJ06-
435. Therefore, after 1855 AD, the sediment of core CJ06-435 mainly included the fine fraction of
the suspended sediment dispersed from the YR mouth, the resuspended sediment from the coastal
area off the YR delta in the winter, and the locally resuspended sediment.
The accumulation of YR-suspended sediment during the summer season in Laizhou Bay was
closely associated with the sediment dispersion pattern off the active delta lobe (Xing et al., 2016).
The estuary of the YR, during most of the period 1855–1976 AD, was north of the modern YRD,
and suspended sediment from the YR was transported northeastward to Bohai Bay and the central
Bohai basin. The contribution of YR-suspended sediment to the sedimentation of core CJ06-435
was smaller and the resuspended sediment became a dominant material source. During the winter,
the large waves generated by the strong winds may result in intensive resuspension of the seabed
sediment and lead to part of the coarse sediment in the YR mouth and Laizhou Bay being transported
to the study area, which induces an evident increase in mean grain size and a decrease in the amount
of fine sediment. After 1976 AD, the lower channel of the YR shifted to the Qingshuigou course in
Laizhou Bay; the suspended sediments derived from the YR estuary were primarily driven
southward and southeastward along the coast, leading an increasing transportation of most of the
YR-suspended sediment into Laizhou Bay (Qiao et al., 2010). As a result, the dispersal of river-
laden sediment contributed substantially to the sedimentation of core CJ06-435, with a fine sediment
layer being formed in the upper part of the core.
The results inferred from our grain-size data on the migrations of the YR lower channel since
1855 AD and their effects on the sedimentary environments of the adjacent BS are in accordance
with the results of other studies from Laizhou Bay (Wu et al., 2015) and central BS (Hu et al., 2011).
Based on the records of sediment core collected from Laizhou Bay, Wu et al. (2015) found that when
the YR mouth approached the core location, the sediment became finer; otherwise, the active
resuspension resulted in the accumulation of coarser sediment owing to strong hydrodynamics. The
grain size results from the central mud areas of the BS also point to the conclusion that the sediment
supply from the YR to the central BS was cut off because of the shift of the YR terminal course
from the Diaokou source in outer Bohai Bay to the Qingshuigou course in Laizhou Bay in 1976;
hence, resuspended sediment became a primary source of sediment dispersal in the central BS. As
a result, there was a significant increase in the proportion of sand in surface sediment in the central
BS (Hu et al., 2011).
It is worth noting that the variation of grain size characteristics in the period of 6570–5000 a
BP is very similar to that after 1855 AD. As shown in Figure 7d, the shift of Mz in the period of
6570–5000 a BP also began with a significantly increased of Mz at 6570 a BP (160 cm) when the
YR flowed into the BS in northern Shangdong province. This similar variation of grain size in the
period of 6570–5000 a BP (superlobe 1) and after 1855 AD (superlobe 10) implies that a similar YR
channel shifting occurred during these two periods. However, further research is needed to reveal
how the deltaic and neritic sea sedimentary environment was impacted by the river system.

**5.3. Coastal salt marsh response to hydrological change**
Two high-amplitude salt marsh vegetation shifts are displayed in the herb pollen record during
6570–5000 a BP (superlobe 1) and after 1855 AD (superlobe 10), indicating rapid oscillations of
environmental conditions in the coastal area of BS. Within single intervals of the YR superlobe, a
recurrent and directional alternation of herb pollen taxa is observed in the following order: the shift
of herb pollen data began with an abrupt decrease of Cyperaceae pollen, followed by a steep increase
of Chenopodiaceae and *Artemisia* pollen (Figure 8b).
Cyperaceae, Chenopodiaceae, and *Artemisia* are the three plant families/genus that contain the
important representatives of coastal salt marsh plants (Lu et al., 2006). In the salt marsh of the
modern YRD, species composition of Cyperaceae, Chenopodiaceae, and *Artemisia* varies with
salinity and soil moisture. Plant families such as Cyperaceae are mainly composed of
hydrophytes and phreatophyte E*leocharis valleculosa*, *Cyperus rotundus*, *Scirpus planiculmis*, *S.*
*triqueter*, *S. yagara*, *S. juncoides*, and *Juncellus serotinus* (Pan and Xu, 2011). The presence of
Cyperaceae necessarily indicates lower saline conditions, since hydrophytes and phreatophyte
sedges typically colonize in the middle and upper part of the supralittoral zone, both sides along the
riverbank, the coast of the lake, and the interfluvial lowlands of the paleo-river. These areas are far
from the coastline, and the main type of soil is salinized soil with lower salinity (Zhang et al., 2009a;
Xu, 2011). Chenopodiaceae are mainly composed of halophyte *Suaeda glauca*, *S. salsa*, and
*Salicornia europaea*. *Artemisia* mainly consist of halophyte and xerophyte *Artemisia carvifolia*, *A.*
*capillaris*, and *A. annua* (Xing et al., 2003; Zhang et al., 2009b). In the modern YRD, halophytes
are distributed in the intertidal zone mudflat and the outside margin part of the supralittoral zone.
These areas are near the coastline, characterized by a high incidence of wave brakes and prolonged
inundation regimes, and the main type of soil is saline (Zhang et al., 2009b). Therefore, in salt marsh
plant communities, the variation in the amount of Cyperaceae, Chenopodiaceae, and *Artemisia* is
often thought to reflect environmental gradients controlled by the distance from the coast, local
topography, terrigenous material, and freshwater input (González and Dupont, 2009; Zhang et al.,
2009a). The pollen record from the BS could provide evidence of coastal salt marsh development
over decades to centuries of the response to environmental alternations during the period of
hydrological change.

In the studied sequence, the YR flowed into the BS toward the northern Shandong Province

after a course shift. The lower river was initially braided upon relocation, as characterized by
unchannelized river flow. At this initial stage, the river-derived sediment was largely accumulated
in the floodplain and/or among the antecedent rivers owing to the lack of channelization (Wu et al.,
2017), filling the coast of the lake, the interfluvial lowlands of the paleo-river, and the supralittoral
zone, etc. This caused the destruction of hydrophytes and phreatophyte sedges in these areas. This
process is indicated in our records by the significant decrease in the amount of Cyperaceae pollen
percentages for superlobe 1 and superlobe 10 (Figure 8b).

Eventually, natural channel adjustments resulted in the coalescence of multiple channels into

a single channel (Wu et al., 2017). A large amount of river-derived sediment was deposited at the
mouth of the YR, causing the progradation of the YRD. Because of the strong influence of the tides,
the intertidal zone in the YRD was originally bare beach. Along with the seaward expansion of the
newly formed beach, the influence of tides was weakening on the original bare beach wetland and
the salinity of the original beach wetland began to decrease (Zhang et al., 2007). Pioneer species of
salt marshes, e.g., *Suaeda glauca*, *S. salsa* and *Salicornia europaea* (Chenopodiaceae), first
colonized this original bare beach (Zhang et al., 2009a). The significant increase of Chenopodiaceae
in our pollen record (Figure 8b) is, therefore, interpreted as the development of the *S. glauca*
population.

**5.4. Palaeovegetation reconstruction and its climate significance**
Based on the age model, the record between 3000 a BP and 1855 AD of core CJ06-435 is
somewhat confused. Because the CSR was extremely low (about 0.005 cm/yr) during 3000 a BP to
1855 AD. As reported in recent studies, the CSR of cores from the tidal flat and neritic sea of the
south BS were in the range 0.02–0.13 cm/yr before 2000 a BP (in cores H9601, H9602, ZK228, and
ZK1, Figure 1; He et al., 2019), 0.04–0.06 cm/yr between 2000 a BP to 1855 AD (in cores ZK228,
HB-1, and GYDY, Figure 1; He et al., 2019), and 0.35–1.38 cm/yr since 1855 AD (Wu et al., 2015;
Qiao et al., 2017; Xu et al., 2018). Although core CJ06-435 is offshore compared to the other cores
(e.g. H9601, H9602, ZK228, HB-1, ZK1 and GYDY), and it should have a lower CSR compared to
those cores. But, the difference of CSR between CJ06-435 and those cores reach up to ten-fold. The
reasonable explanation is that there might be some deposition hiatus between 3000 a BP and 1855
AD of core CJ06-435. The calculated CSR in the upper layer (since 1855 AD, as calculated to 0.17–
0.48 cm/yr) and the lower layer (3000–8500 a BP, as calculated to 0.016–0.057 cm/yr) of core CJ06-
435 are comparable to the nearby records by He et al. (2019) and Xu et al. (2018). Therefore, we
just focused on the vegetation successions and climate change between 8500 and 3000 a BP, and
only gave a cautious discussion for the chronology uncertain interval in this study.
During the period from 8500 to 6500 a BP (palynological zone 1, 271–156 cm), the
palynofloral assemblages are mainly composed of the pollen of broadleaved trees, such as *Quercus,*
*Betula*, *Alnus*, and Ulmaceae, combined with the pollen of hydrophytes and phreatophyte
Cyperaceae and *Typha*; of these, the pollen of *Quercus* and *Typha* are predominant (Figures 4 and
5). In contrast, the pollen of halophytic and xerophytic herbs and conifer trees is scarce. The pollen
assemblages encountered herein indicated that the vegetation of the BS land area consisted mainly
of oak-rich temperate broadleaf deciduous forest, with some conifer trees on the uplands, and
freshwater lakes and marshes dominating the coastal area, under the influence of a markedly warmer
and wetter climate than the present. The highest values of AP pollen concentration (Figure 5),
reflecting a dense vegetation cover, also represent warm conditions during this period. The pollen
data are comparable to that found from previous palynological studies carried out in north China
(e.g., Yi et al., 2003; Ren, 2007; Chen and Wang, 2012; Li et al., 2019) and northeast China (e.g.,
Ren and Beug, 2002; Li et al., 2011; Stebich et al., 2015), from which a warm, wet climate
corresponding to the Holocene Optimum was inferred. Under the influence of the Holocene
Optimum, the forest cover evidently increased in north and northeast China (Ren, 2007). In the YR
drainage area and Shandong Peninsula, the broadleaved deciduous forest thrived, accompanied by
the presence of monsoonal evergreen forest and the abrupt decrease in the herbaceous taxa and
conifers (Yi et al., 2003; Chen and Wang, 2012; Li et al., 2019).
During the period from 6500–5900 a BP (palynological zone 2a, 156–128 cm), a climatic
cooling period is identified by an increase of conifers, *Pinus*, combined with an abrupt reduction of
broadleaved trees (*Quercus*, *Betula*, *Alnus*, *Pterocarya*, Ulmaceae, and Moraceae). Halophytic and
xerophytic herbs taxa such as Compositae, *Artemisia*, and Chenopodiaceae also increase, while
Cyperaceae and aquatic herbs *Typha* obviously decreased. The climate shifted from warm, wet to
cool, and dry may have caused the reduction of broadleaved deciduous forests, the expansion of
conifer forests, and the gradually disappearance of freshwater lakes and marshes that had spread
over the coastal area of BS. This result is in good agreement with previous studies. A pollen record
from Shandong Peninsula revealed that *Quercus* content decreased, and herbs increased quickly
following the Holocene Optimum, indicating a potential climate deterioration (Chen and Wang,
2012). A pollen record from Lake Bayanchagan in southern Inner Mongolia also showed that
deciduous trees declined, conifers reached their maximum values, whilst steppe vegetation remained
relatively high during 6500–5100 a BP, indicating cold and dry climate conditions (Jiang et al.,
2006). It is worth noting that, a vast delta complex began to build up in the western part of the BS
after 6570 a BP, which resulted in the increase of land area and development of the YR delta wetland.
It can be concluded that the expansion of salt marsh during this period may be partly related to the
formation of the YR delta complex.
During the period from 5900–3500 a BP (palynological zone 2b, 128–63 cm), climate cooling
and drying is observed by a reduction of broadleaved trees such as *Quercus*, *Pterocarya*, Ulmaceae
and Moraceae, and a rising frequency of halophytic and xerophytic herbs (*Artemisia* and
Chenopodiaceae) (Figures 4 and 5). The cool, dry conditions probably cause the contraction of
broadleaved forest and the expansion of halophytic and xerophytic herbs, which is similar to the
findings of a study by Jiang et al. (2006). Based on the quantitative climatic reconstruction from
pollen and algal data for Lake Bayanchagan, Jiang et al. (2006) found that broadleaved trees, such
as *Betula*, *Corylus*, *Ostryopsis* and *Ulmus* further declined, whereas the amount of steppe vegetation
increased. The reconstruction of mean annual temperature and total annual precipitation dropped to
their minimum values during 5100–2600 a BP.

Between 3500 and 3000 a BP (63–56 cm), a warm climatic phase occurred, as suggested by an

increase in the amount of pollen from broadleaved trees (*Quercus*, *Betula*, *Alnus*, *Pterocarya*, and
Ulmaceae), with low frequencies of conifer pollen (*Pinus*). Moreover, halophytic and xerophytic
herb pollen, including Compositae, *Artemisia*, and Chenopodiaceae, were still present at a high
frequency (Figures 4 and 5). Accordingly, the assemblages reveal that a warm, dry climate probably
developed during this period.

From 3000 a BP to 1855 AD (56–30 cm), as mentioned in the first paragraph of this section,

the CSR was extremely low during the period from 3000 a BP to 1855 AD (56–30 cm). We suggest
that there might be some deposition hiatus and only tentatively discuss this section of the pollen
record. This section begins with a relatively high percentage of broadleaved trees (*Quercus*, *Betula*,
*Alnus*, *Pterocarya*, and Ulmaceae) and low frequencies of conifer pollen (*Pinus*) (56-41 cm); this is
consistent with the previous stage (3500–3000 a BP, 63-56 cm). Afterward, there is a dramatic
decrease in the occurrence of *Quercus*, quickly followed by a sudden increase in *Pinus* (41–30 cm,
Figures 4 and 5). These pollen data suggest that the climate was warm during the latter part of this
period (56–41 cm, 3000 a BP–?), following the climate condition of previous stage (63–56 cm,
3500–3000 a BP). In the earlier part of this period (41–30 cm, ?–1855 AD), the sudden increase in
*Pinus* and major reductions of *Quercus* are likely signs of human impacts on the natural vegatation,
including deforestation and cultivation. Park and Kim (2015) interpreted the decrease in the
percentage of *Quercus* and increase of *Pinus* in the late Holocene as marking the development of
secondary vegetation under anthropogenic influence. Based on two boreholes palynological from
the YRD, Yi et al. (2003) found a sudden reduction of *Quercus*, followed by a marked increase of
*Pinus* after 1300 a BP. Their research considered that this typical lag between the two taxa may
indicate that, after the clearance of the local broadleaved deciduous forests, the vegetation was
replaced by a secondary pine forest.

After 1855 AD (palynological zone 3, 30–0 cm), a significant decline in broadleaved trees

(*Quercus*) and conifers (*Pinus*) pollen, as well as an increase in the frequency of herbs (Poaceae,
Compositae, *Artemisia*, and Chenopodiaceae) pollen may reflect the further strengthening of human
disturbance on the vegetation and the expansion of intensive agricultural cultivation into forests of
the BS coastal area. Moreover, after 1855 AD, the present YR began returning to the BS, and
forming a vast area of floodplain and estuarine wetland on the southwest coast of the BS (Saito et
al., 2000; Jiang et al., 2013). The variation of herb pollen may be partly related to the development
of the modern YRD wetland.

**5.5. Holocene temperature variations in north China and possible driving mechanisms**
Many previous studies of north and northeast China have used the *Quercus* pollen percentage
to infer regional temperature variation (Ren and Zhang, 1998; Yi et al., 2003; Li et al., 2004; Xu et
al., 2014; Zhang et al., 2019). The *Quercus* pollen percentage from CJ06-435 core is consistent with
previous studies, which also provide a regional air temperature reference. As shown in Figure 9d,
the percentage of *Quercus* pollen in CJ06-435 core indicates a warm early Holocene from 8500 to
6500 a BP, a cool mid-Holocene from 6500 to 3500 a BP, and then a relatively warm late Holocene.
These climate changes were also apparent in the change of *Quercus*/*Pinus* (Q/P) ratio. The average
Q/P ratio was approximately 0.57 between 8500 and 6500 a BP, and changed to 0.33 between 6500
and 3500 a BP, and then gradually increased (Figure 9c).
The pollen-based record of temperature evolution from core CJ06-435 are broadly in-phase
with published high-resolution sea surface temperature record from core YS01 (Figure 9e) of the
Yellow Sea, suggesting at least a local pattern of temperature variations during the Holocene. To
investigate whether the temperature pattern was local characteristics of the BS and Yellow Sea area,
or whether it was rather a regional patterns of East Asia as a whole, the pollen records of core CJ06-
435 were compared with recently-published and relatively well-dated sequences from north China,
northwest China, and the Tibetan Plateau (see Figure 9a for site locations), including the
sedimentary pollen-based temperature record from the Narenxia Peat within the Kanas Lake,
Northwest China (Figure 9f; Feng et al., 2017); $U^{K}_{37}$ inferred temperature record at Lake Qinghai
(Figure 9g; Hou et al., 2016); and lacustrine sedimentary pollen-based quantitative temperature
record (the mean annual temperature) from Lake Bayanchagan in Inner Mongolia in north China
(Figure 9h; Jiang et al., 2006). All three records indicate that the temperature was high between

8500 and 6000 a BP, low between 6000 and 4000–3000 a BP, and averagely high after 4000–3000 a BP (Figure 9f–h), this is consistent with the basic change pattern of our pollen-based temperature variation (Figure 9c and 9d). The comparison of the five records demonstrates that the CJ06-435 core *Quercus* pollen percentage record is, at a minimum, of regional significance.

Insolation has been widely accepted as an important factor in Holocene climate variation. The covariation of Northern Hemisphere extratropical (30 ° to 90 °N) temperature and local summer insolation on an orbital scale, and the long-term decrease of summer insolation make the especially pronounced cooling of the Northern Hemisphere extra-tropics during the Holocene (Marcott et al., 2013) appear reasonable. However, the general pattern of temperature variation revealed by our study is not entirely consistent with local mean annual insolation forcing (Figure 9b). Our results indicated a cold mid-Holocene from 6500 to 3500 a BP and a relatively warm late Holocene. This temperature characterization of a cool mid-Holocene and a relatively warm late Holocene is also seen in many proxy records in the East Asia (Thompson et al., 1997; Jiang et al., 2006; Hou et al., 2016; Wu et al., 2016; Feng et al., 2017; Jia et al., 2019). The cooler mid-Holocene seen in the East Asia, could not solely be explained by the gradually decreasing summer insolation during the Holocene but might be related to other forcings.

We compared the pollen-based temperature record of core CJ06-435 (Figure 9c and 9d) with the frequency of El Niño events reconstructed from the Botryococcene concentration in the El Junco Lake sediment (Figure 9i; Zhang et al., 2014) and the ENSO variability reconstructed from $\delta^{18}O$ values of individual planktonic foraminifera retrieved from deep-sea sediments (Figure 9j; Koutavas and Joanides, 2012). As shown in Figure 9, lower temperature periods in the mid-Holocene tend to occur during a period of low El Niño activity, a relatively high temperature period in late Holocene tend to occur during a period of high El Niño activity, which indicates that there may be some link between the temperature of BS and Yellow Sea area and the ENSO system. Modern research suggests that ENSO can influence the evolution of temperature behavior, over interannual to multi-decadal time ranges (Hoerling et al., 2008; Triacca et al., 2014). In East Asia, many studies have indicated that the East Asian winter monsoon and the ENSO are tightly coupled (Zhou et al., 2007; Cheung et al., 2012; An et al., 2017). Generally, the strength of winter monsoon and East Asia troughs weakens in an El Niño year, and the weakening could cause the observed winter half-year warming (Xu et al., 2005). On centennial/millennial time scales, using pollen data from Lake Moon

in the central part of the Great Khingan Mountain Range, Wu et al. (2019) recently connected increased El Niño frequency with the decrease of winter monsoon activity in the East Asia, and the warming winter temperature in the Great Khingan Mountain Range since the mid-Holocene. Feng et al. (2017) founded that warm-phased ENSO was teleconnected with weakening of the Siberian High, and that the weakening was a cause of the observed winter half-year warming in southern Siberia. Likewise, the results of this study, indicating more or less synchronicity of the climate change in north China and ENSO activity, provide a possible linkage between the climate of north China and oceanic forcing during the mid-late Holocene.

In addition, radiative forcing by greenhouse gases (GHGs) rose 0.5 $W/m^2$ during the mid-late Holocene (Marcott et al., 2013), which would be expected to yield 1°C warming at Kinderlinskaya Cave in the southern Ural Mountains from 7000 a BP to the pre-industrial (Baker et al., 2017). Recently, both winter insolation and GHG forcing have been proposed as the major driving factors for winter warming during the Holocene in the Siberian Arctic (Meyer et al., 2015) and the southern Ural Mountains (Baker et al., 2017). Similarly, summer warming in central Asia during the mid-late Holocene, recorded by the alpine peat α-cellulose $\delta^{13}C$ record from the Altai Mountains (Rao et al., 2019), has been proposed to be mainly driven by the enhanced GHG forcing and increasing human activities. Rao et al. (2020) suggested that GHG forcing was the dominant driver of the summer and winter warming trends since ~5000 a BP. The effects of GHG forcing are global. Hence, we suggest that enhanced GHG forcing may be an important driver for mid-late Holocene temperature variations of East Asia.

In summary, the temperature characterizations of a cool mid-Holocene and a relatively warm late-Holocene revealed by the East Asia records could be linked with the change of insolation, ENSO activity and GHG forcing. The cooler mid-Holocene may be related to a combination of the decreasing summer insolation, weak El Niño activity, and relatively lower GHG radiative forcing during this interval. Along with strengthened ENSO activity and enhanced GHG forcing in the late Holocene, there was increased temperature.

## 6. Conclusions

Through the palynological and grain size reconstruction of coastal area vegetation and environment in core CJ06-435, we were able to identify specific responses of plant communities to

climatic (temperature, precipitation), hydrological, and anthropogenic impacts. Our data elucidate the pattern and mechanisms driving coastal salt marsh succession at decade-to-century timescales. Two intervals of expanded salt marsh vegetation correspond to the formation of YR delta superlobes, indicating that soil development and salinity gradients are the main factors determining the vegetation dynamics of coastal wetland. Our pollen-based temperature index revealed a warm early Holocene (8500–6500 a BP), a subsequent cool stage between 6500 and 3500 a BP, and a slightly warming episode after 3500 a BP. The reliability of the record, especially the cooler mid-Holocene, is further supported by several other temperature records from East Asia. We suggest that changes in insolation, ENSO activity and GHG forcing could have played an important role in the temperature evolution at the East Asia.

**Code/Data availability**

The co-authors declare that all data included in this study are available upon request by contact with the corresponding author (Email: *jinxiachen@fio.org.cn*).

**Author contribution**

Chen Jinxia wrote the manuscript; Shi Xuefa and Liu Yanguang revised the manuscript; Qiao Shuqing provided many constructive suggestions for the manuscript; Yang Shixiong provided the pollen data of surface sediment; Li Jianyong use pollen data of core CJ06-435 as base for quantitative climate reconstruction; Yan Shijuan, Lv Huahua, Li Xiaoyan and Li Chaoxin provided financial support for the collection of samples and obtained samples.

**Competing interests**

The authors declare that they have no conflict of interest.

**Acknowledgements**

We thank the crew of the R/V *Kan407* for sampling. We also thank Dr. Nan Qingyun for improving this paper. This work was supported by the National Natural Science Foundation of China (Grant Nos. 41420104005 and 41576054).

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

**Table captions**

**Table 1**
AMS radiocarbon dates from core CJ06-435 and one tie points corresponding to the deepest onset of
$^{137}$Cs in environmental samples at measurable levels; for calibration in years before present (a BP) 0 =
1950 AD.

| Core depth (cm) | Materials | Radiocarbon date (a) | Age error (a) | Calibrated age (1σ) (a BP) | Mean calibrated age (a BP) | Laboratory |
|---|---|---|---|---|---|---|
| 25 | $^{137}$Cs | – | – | – | –4 | NIGLAS |
| 7 | Mixed benthic foraminifera | 3020 | 30 | 2854-3039 | 2951 | Beta |
| 13 | Mixed benthic foraminifera | 2990 | 30 | 2817-2997 | 2913 | Beta |
| 17 | Mixed benthic foraminifera | 3060 | 30 | 2908-3102 | 3003 | WHOI |
| 59 | Mixed benthic foraminifera | 3340 | 30 | 3270-3485 | 3359 | Beta |
| 69 | Mixed benthic foraminifera | 3590 | 25 | 3563-3725 | 3656 | WHOI |
| 87 | Mixed benthic foraminifera | 4450 | 30 | 4695-4878 | 4801 | Beta |
| 119 | Mixed benthic foraminifera | 5200 | 30 | 5604-5770 | 5706 | WHOI |
| 129 | Mixed benthic foraminifera | 4520 | 30 | 4812-4965 | 4894 | Beta |
| 161 | Mixed benthic foraminifera | 6020 | 30 | 6501-6667 | 6592 | WHOI |
| 183 | Mixed benthic foraminifera | 6340 | 35 | 6886-7081 | 6981 | WHOI |


















**Figure caption**

**Figure 1:** Geographic map of the Bohai Sea, with locations of core CJ06-435 (red circle) and other sites referred to in this study (purple circles). Cores references: H9601 and H9602 (Saito et al., 2000), ZK1 (Li et al., 2013), ZK228 (Xue et al.,1988), HB-1 (Liu et al., 2009a), GYDY (Liu et al., 2014).

**Figure 2:** Vegetation map around the Bohai Sea and the ocean current in the Bohai Sea during the summer (a) and winter (b) (YSWC: Yellow Sea Warm Current; SBSCC: Southern Bohai Sea Coastal Current; LNCC: Liaonan Coastal Current; LBCC: Lubei Coastal Current, modified from Qiao et al., 2010; the vegetation dataset was provided by the Environmental and Ecological Science Data Center for West China, National Natural Science Foundation of China [http://westdc.westgis.ac.cn] and is based on the Vegetation Atlas of China, 1:1000,000; Hou, 2001).

**Figure 3:** Lithology, grain size, color reflectance $a$*, magnetic susceptibility, and activity profiles for $^{137}$Cs and $^{210}$Pb of core CJ06-435.

**Figure 4:** Percentage diagram of the principal pollen taxa from core CJ06-435. Pollen zonation is based on CONISS results.

**Figure 5:** Concentration diagram of the principal pollen taxa from core CJ06-435.

**Figure 6:** Spatial distribution of modern pollen percentage (black solid circle, %) and concentration (red open circle, grains/g) in Laizhou Bay, Bohai Sea (modified from Yang et al., 2016).

**Figure 7:** (a-f) Vertical profiles of grain-size parameters and halophytic and xerophytic herb (Chenopodiaceae and *Artemisia*) pollen percentage and concentration of core CJ06-435 (Mz – mean grain size). (g) The location of Yellow River superlobe 1 (Lijin superlobe) and superlobe 10 (Modern superlobe) (modified after Xue, 1993).

**Figure 8:** (a) Correlating proxy to paleo-superlobe variation of the YR, from top to bottom: percentage of Cyperaceae, Chenopodiaceae, and *Artemisia* pollen; concentration of Chenopodiaceae, *Artemisia*, and Cyperaceae pollen; sand percentage. (b) Detailed pollen and grain size profiles representing salt marsh species (Cyperaceae, Chenopodiaceae, *Artemisia*) relative abundances and hydrodynamic change during the formation of Yellow River superlobe 1 and 10. Pollen percentage of Cyperaceae, Chenopodiaceae and *Artemisia* from core CJ06-435 indicating the directional alternation of salt marshes along the Bohai Sea, ①— Unchannelized river flow characterized the onset of Yellow River channel shift, caused a large amount of river–derived sediment accumulation in the floodplain and destroyed the sedges in the coastal depression; ②

—Along with the formation of a new channel, lateral migration of the lower channel stopped, and
new intertidal mudflat was formed. Pioneer species (Chenopodiaceae, *Artemisia*) first colonize
bare zones of lower and middle marsh areas.


**Figure 9:** Comparison of relevant Holocene temperature records with solar irradiance and ENSO
proxy records derived from the equatorial Pacific. (a) Locations of the sites where the Holocene
temperature records are derived. The schematic large-scale diagram modified from Hao et al.
(2017). In the diagram, the purple area is Tibetan Plateau, the yellow area is Chinese Loess Plateau,
and red circle refer to the corresponding study sites. (b) Summer (mean of June) insolation
irradiance for the Northern Hemisphere (40°N). (c,d) *Quercus*/*Pinus* (Q/P) rate and *Quercus* pollen
percentage records from core CJ06-435, bold-blue and bold-red lines are the five-point running
average. As introduced in part 5.4 "Palaeovegetation reconstruction and its climate significance",
there might be some deposition hiatus between 3000 a BP and 1855 AD. So, Q/P rate and *Quercus*
pollen percentage records during 3000 a BP and 1855 AD is for reference only. (e) $U^{K'}_{37}$ SST record
from YS01 core in the south Yellow Sea (Jia et al., 2019). (f) Pollen-based mean annual temperature
(MAT) record from Narenxia Peat in the southern Altai (Feng et al., 2017). (g) $U^{K}_{37}$ inferred
temperature record at Lake Qinghai (Hou et al., 2016). (h) Pollen-based mean annual temperature
record from Lake Bayanchagan in Inner Mongolia, North China (Jiang et al., 2006). (i)
Botryococcene concentrations in the El Junco sediment, a proxy for frequency of El Niño events
(Zhang et al., 2014). (j) Variance of $\delta^{18}O$ values of individual planktonic foraminifera (*G. ruber*) in
sediment core V21–30 from the Galápagos region, a proxy for ENSO variance (Koutavas and
Joanides, 2012).










Figure 1

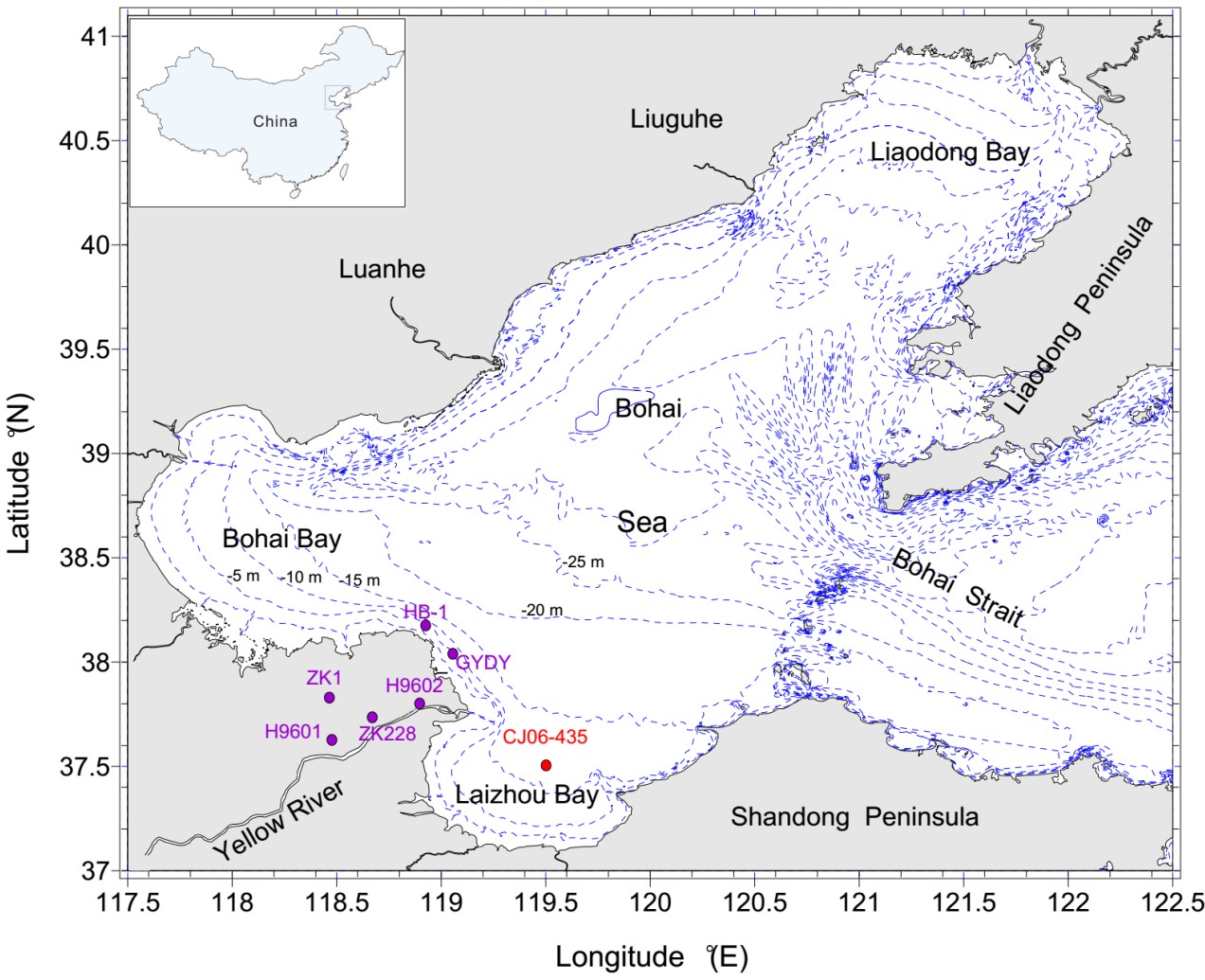






**Figure 2**

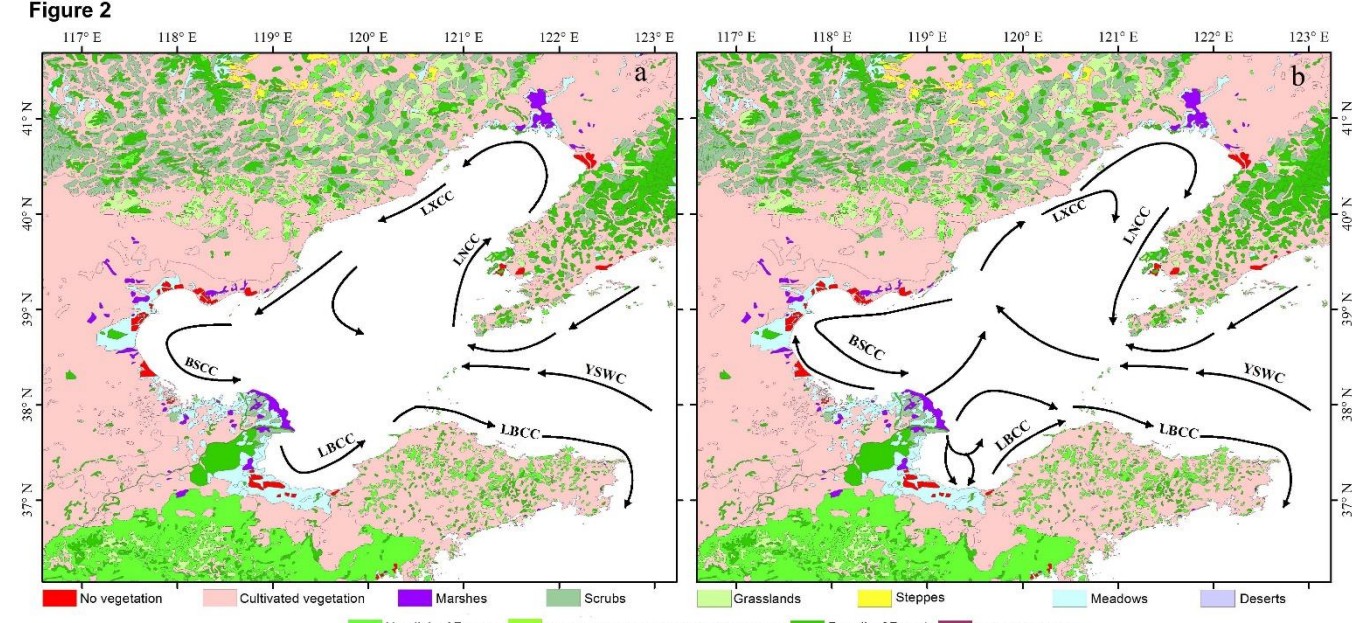

Figure 3

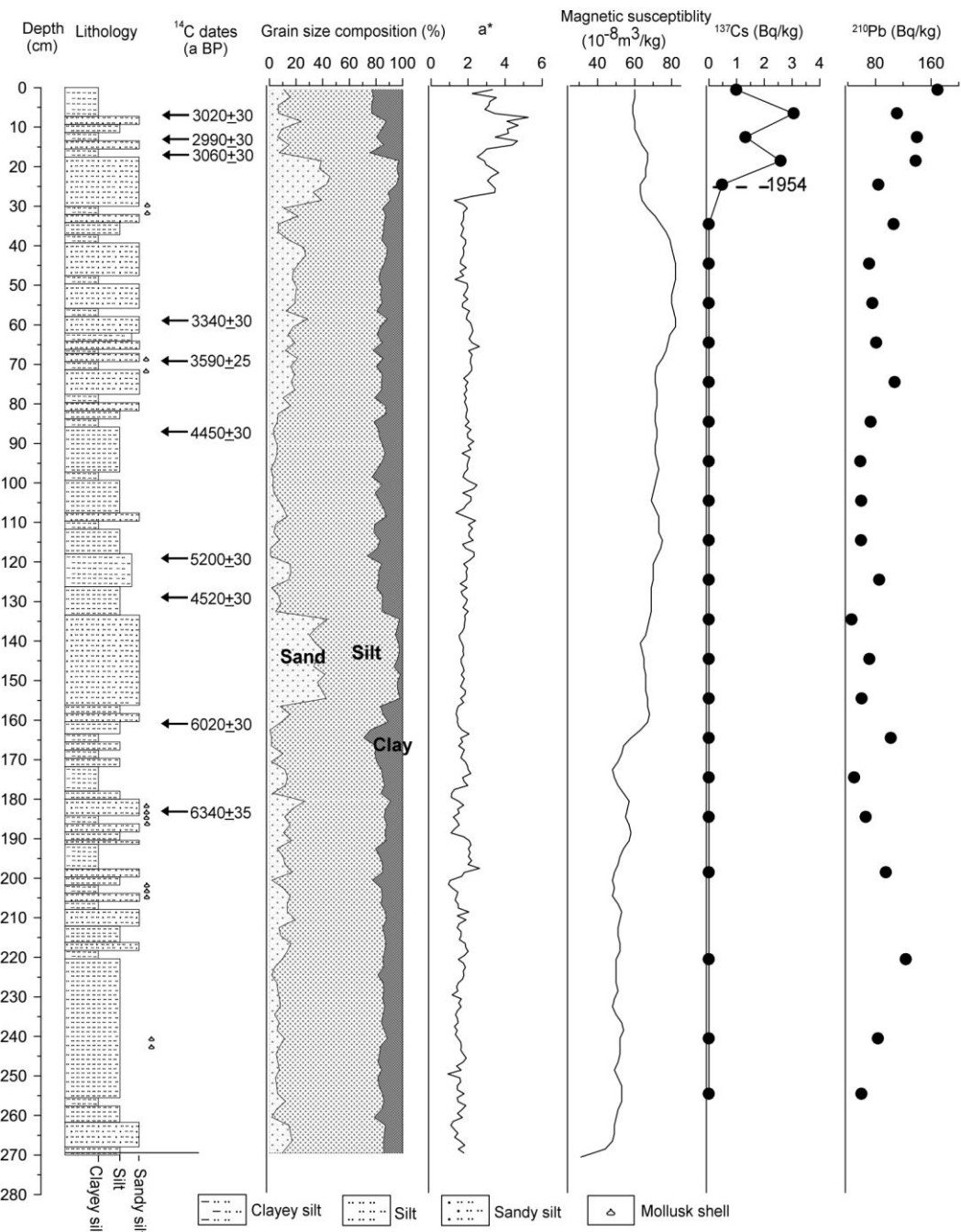


Figure 4

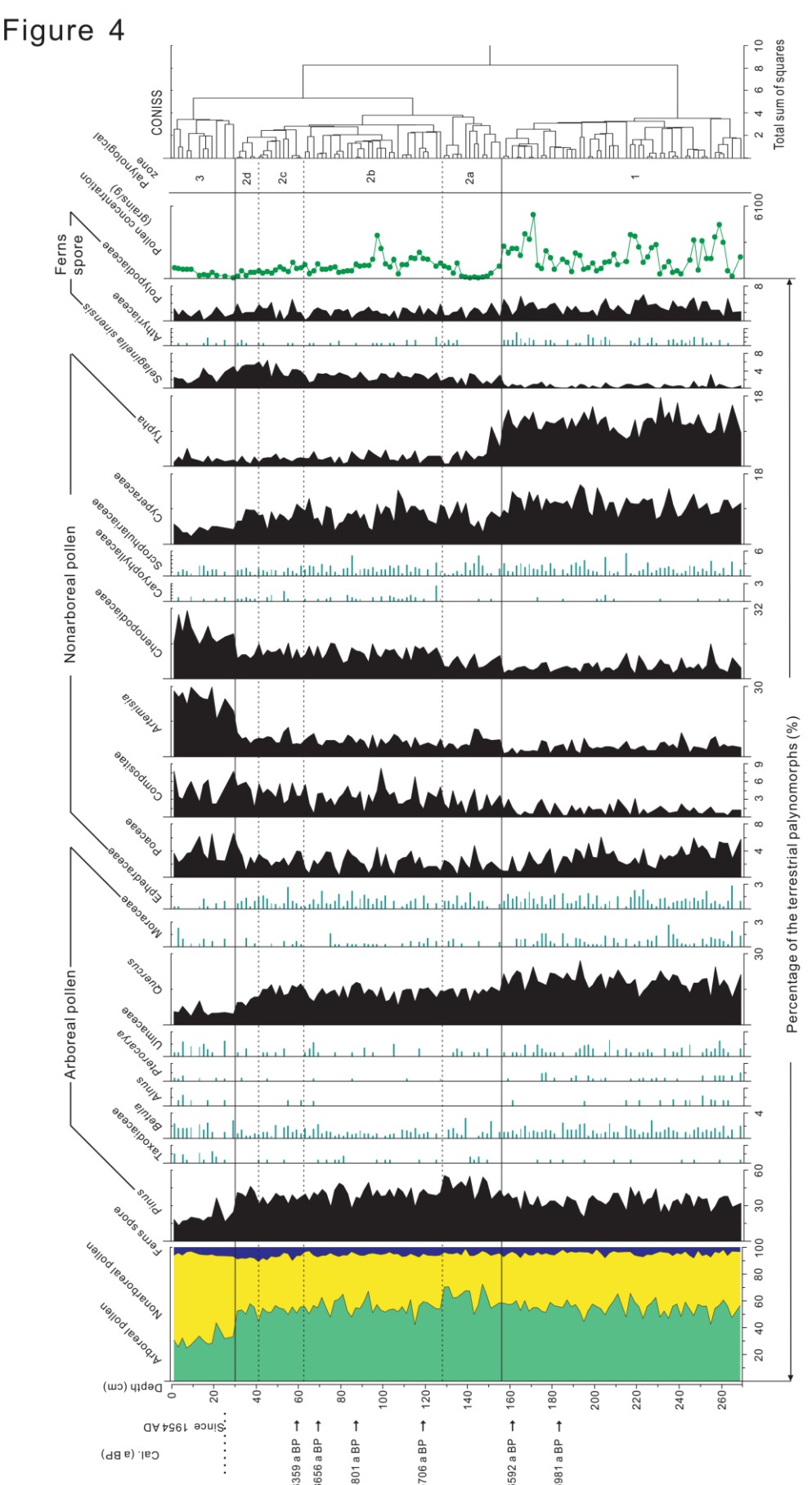


Figure 5

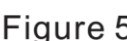


Figure 6

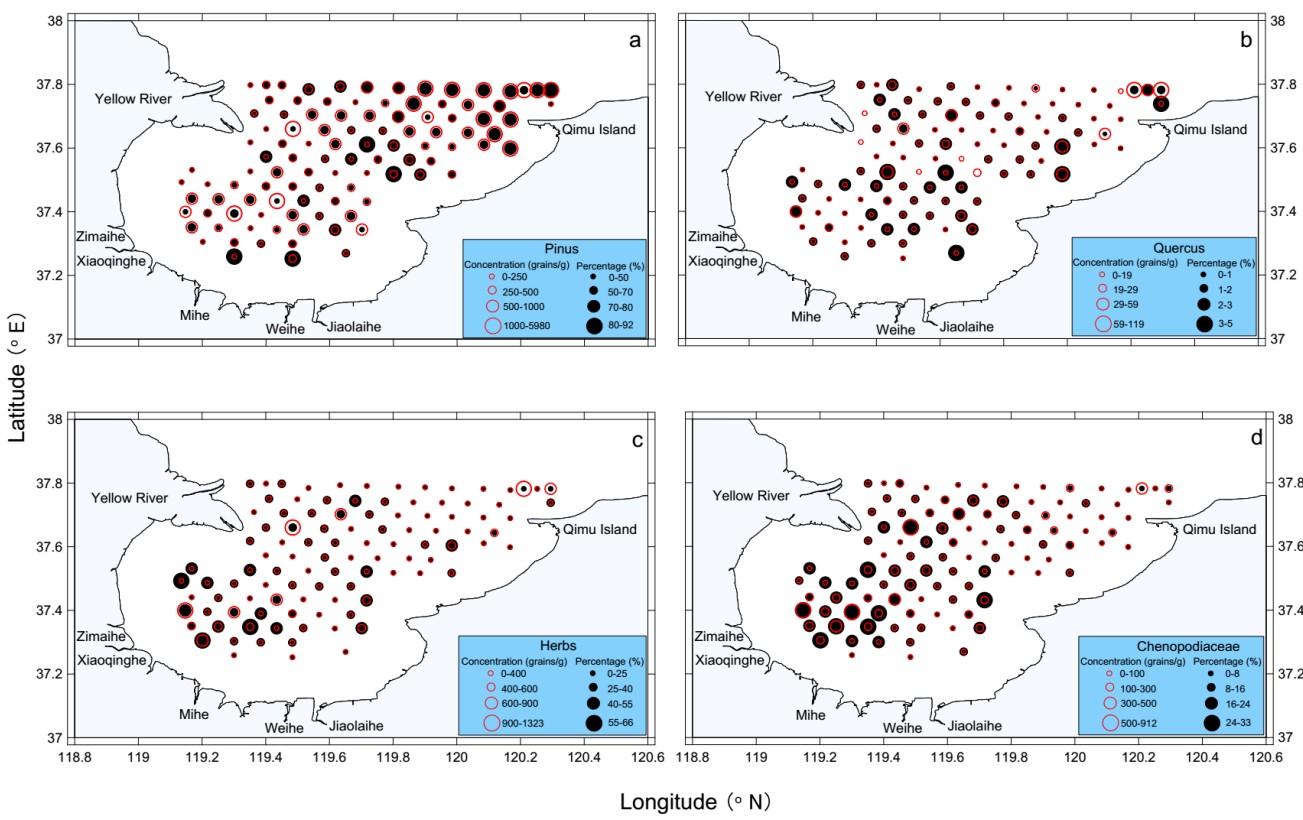











Figure 7

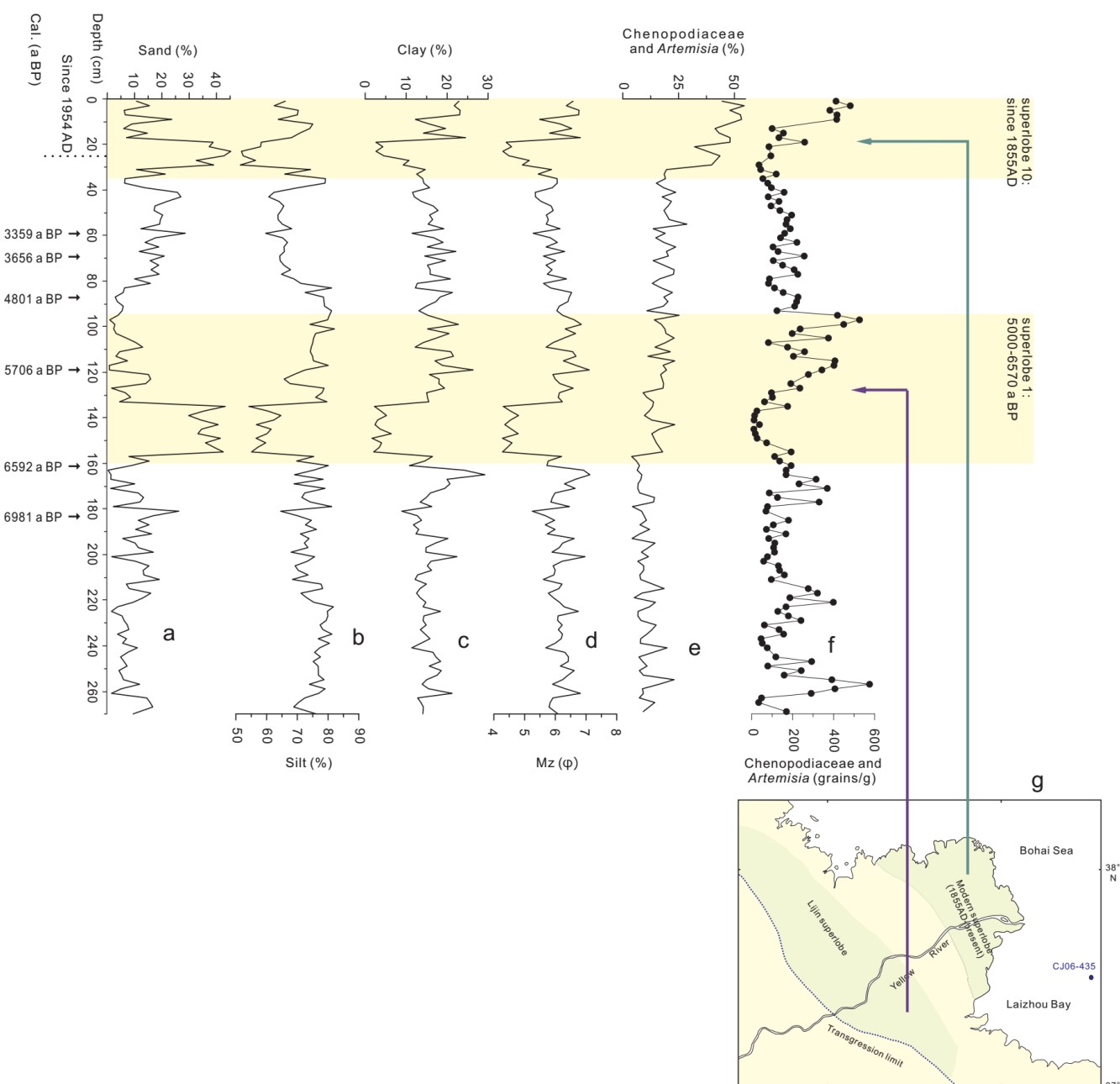



Figure 8

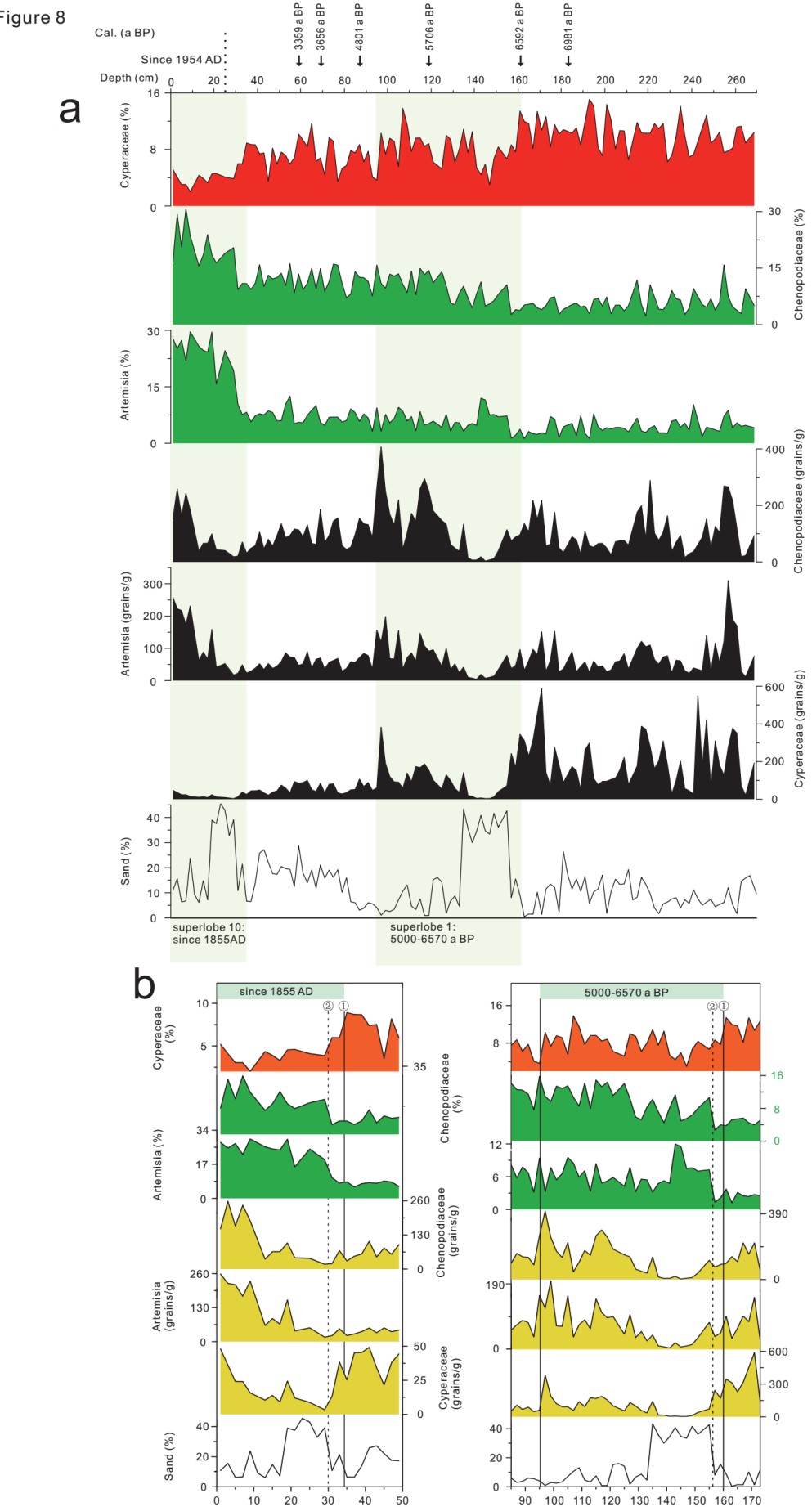





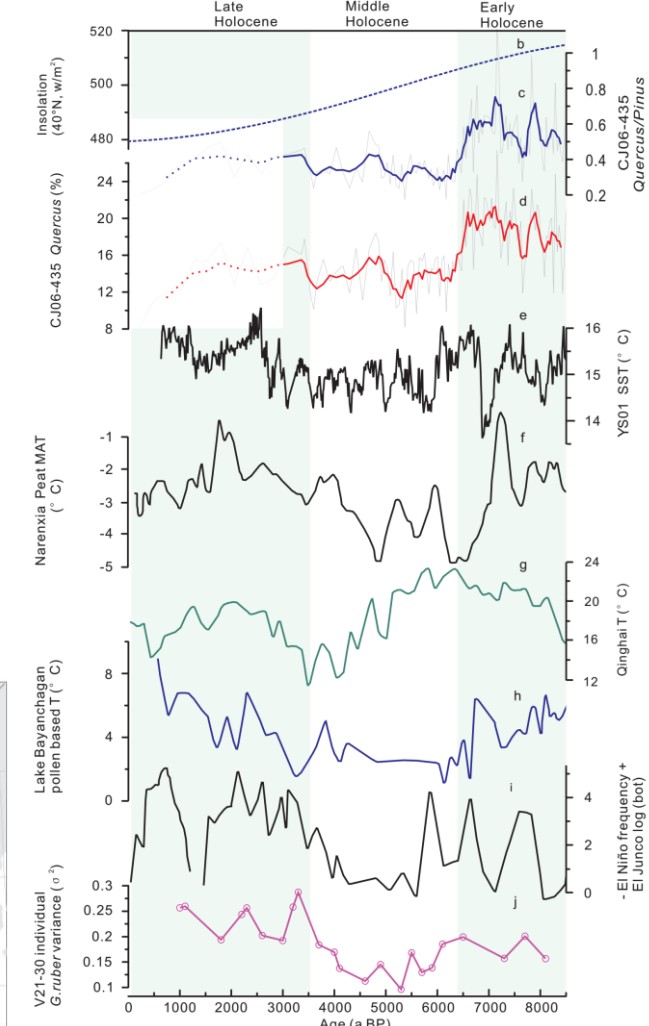