# Peer review of "in the Bohai region"

_Climate of the Past, 2020_

## Referee Comment (RC1) · Anonymous Referee #1 · 20 Apr 2020

This manuscript (MS) by Jinxia et al. describes the results of a palynological and sedimentological study based on a core taken from the Laizhou Bay in the Bohai Sea, eastern China. The MS is generally well written (some remarks concerning grammar etc. and also concerning redundancies are mentioned later), and I think the results are worth being presented since they are at least of regional importance. I am not completely sure though if the results are completely fitting with the scope of Climate of the Past. I have marked the scientific significance as 'good', but in the current stage it is rather between good and fair - if additional methods were used to generate climate data via the described record and if certain aspects would be discussed in more detail, 'good' would probably be fitting.

[Figure]

I must admit in this context that I am not familiar with the research region. It seems some of the authors have also contributed to additional studies from the Bohai region, and I was concerned that there might be some redundancy to these different studies. It seems though that a medium- to high resolution study comprising the Holocene has not yet been published for the region, and the earlier studies are incorporated into the discussion. Several palynological aspects are well incorporated into the manuscript, but a potential degradation of pollen grains is not appropriately discussed in my opinion. The authors correctly refer to the possible problem that there may be a transportation bias concerning bisaccate pollen (it should be considered that this may also affect Poaceae and perhaps Cyperaceae values). They also mention that the suspension of pollen grains may play an impotant role. But pollen concentration varies between less than 100 grains/g (which is quite low!) and several thousand grains/g in this record – this may also point to a degradation signal. Just for example, note that Pinus pollen, which is probably much more resistant to degradation than certain nonsaccate pollen such as Quercus or Chenopodiaceae (compare e.g. Cheddai & Rossignol-Strick 1995 or Havinga 1967), increases relatively in those sections of the core which are characterized by higher sand content (and thus probably are better oxiginated). I think the aspect of pollen preservation should be mentioned in detail at the onset of the discussion and considered throughout.

Another weakness is the age model, particularly in the upper part of the core. The authors dismiss three foraminifer-based ages in favor of one Cs age in the upper section of the core, which may be okay – but then it should be discussed in detail what may have caused that these ages are ca. 3000 yr too old, and how the reader can be sure that the other foraminifer-based ages are correct. It should be mentioned, considering the problems with the uppermost ages, which foraminifer species were chosen (probably benthics?). If the used ages are correct, the 35-cm-section between 25 and 60 cm comprises more than 3000 yr, but the lower 120 cm comprise also ca. 3000 yr. This is certainly tied to the sedimentological aspects which are discussed (particularly the shift(s) of the the Yellow River Channel) by the authors, but in my opinion there remain

a lot of uncertainties concerning the ages particularly above 60 cm depth. Therefore, it seems problematic to me to mention quite precise ages for the uppermost 60 cm (as done e.g. in the abstract, see below). Consider also a possible problem in Tab. 1 (see below). And how have you dealt with the older age at 119 cm compared to the younger at 129 cm – could it be redeposition? Was one age excluded?

I cannot say much about the sedimentological interpretation. Concerning the palynology-related sections of the interpretation, I think they are quite well written (though the preservation and age model 'problems' should be considered more often, particularly when it comes to the interpretation of stages 2c to 3). I wondered in this context if the results of Li et al. 2019 would be worth being mentioned here since the record presented in the MS seems to cover the end of the Holocene climatic optimum.

What concerns me particularly when it comes to the relevance of this MS for Climate of the past is section 5.5, in which the authors link their own date to other climate records.

A) Since the authors show their own data vs age (Fig. 8), it should be clearly stated how the age model was composed (linear interpolation?), it is not enough to show the dates in Tab. 1. What a about the ages at 119 and 129 cm?

B) Quercus, as the authors explain, has several species in the region, therefore, the climatic susceptibility of this genus might be relatively low, and several factors, not only temperature, are influencing its relative occurrences in the pollen record. Quercus pollen is also quite susceptible to taphonomical bias (s.a.). For example, the Quercus curve of the Feng et al. (2017) record the authors cite is (naturally) completely different.

C) Two other studies with pollen-based climate reconstructions which are included in Fig. 8 work with quantitative reconstructions – if the authors want their own data to be directly comparable, they should also use such an approach.

The whole section 5.5 seems a little bit like an addition to make the paper a 'climate paper". This is also consistent with some inconsistencies concerning the related Fig.

8 (see below). In order to make this MS appropriate for Climate of the Past, I would suggest to use the pollen data as base for quantitative climate data. The results should be incorporated in the climate-related section. The other aspects I mentioned (taphonomy/degradation and discussion of the age model/interpolation) should be considered, too, and discussed appropriately.

Some detailed remarks follow below:

Abstract

LINES 14/15: 'Nevertheless... remain sparse.' This sentence implies that this is generally the case, but there are numerous studies from other regions regarding this aspect. Also 'long-term' may be confusing here since the presented record does not even span the whole Holocene. Sentences like this one might perhaps be completely removed.

LINE 29 and following: If I did not completely miss anything, the age model is quite unsure between 3000 and 0 years BP (see general comments), and Pinus (excluding on peak that might be a taphonomical signal, s.a.) seems to be decreasing, compare authors' own results (4.2.3).

LINE 30: I understand this that way that the authors call the Quercus percentages a 'temperature index', which is very keen!

1. Introduction LINES 59/60 While the abbreviations YR and BS are already explained in the abstract, maybe they should be explained again in the main text?

2.2. Climate and vegetation LINE 103 '... annual mean air temperature is 7.5-14.0 °C...' Quite a wide range for an average temperature. LINE 109 Perhaps 'Quercus dentata'?

3.2. Palynological and grain size sample analysis LINE 133: Lycopodium in italics LINE 134: Since KOH also degrades pollen, it should be mentioned how long it was used and if all samples were exposed for an identical time interval. LINE 137: 'palynomorph

sum' – is the pollen sum meant? If also dinocysts and other palynomorphs have been counted, this should be mentioned here. LINE 137: 'exotic pollen method. . .' The whole sentence seems a little queer to me, and if Lycopod spores were used, I find the term 'pollen' misleading in this context.

4.1. Chronological model LINE 159: and following: It should be explained which objects were used for the dating (ideally the specific species should be mentioned). Either here or in the discussion it should be discussed what may have caused the discrepancies and why the authors trust the other AMS radiocarbon dates.

4.2.1 Palynological Zone 1 LINE 174: 'pollen' is a singular tantum, a plural may only be appropriate if one mentions different pollen types (but even then, 'pollens' should better be avoided – occurs again later in the MS). Line 178: The MS should be consistent concerning grains/g and grains g-1.

4.2.2 Palynological Zone 2 LINE 186: 'decline' (Plural) LINE 193: Here, NAPs may be appropriate, but I would still suggest to write NAP. LINE 197: 'percentage frequency' sounds/reads strange – percentage implies relative frequency. . . (occurs again later in the MS)

5.1. Key terrestrial. . . LINE 237: The second sentence sounds/reads strange, and phrases like 'It is worth noting' should be avoided – if it was not worth noting, why should one mention it. LINE 238: There have been many earlier studies which revealed this effect. Perhaps it would be good to add ', also for Asian regions' or something similar after 'Previous studies', or you should cite one older study dealing with the effect. LINE 257: The last sentence seems useless to me.

5.2. Sedimentary records. . . LINE 279: I think these sentences can be significantly condensed. And in this paragraph, the aspect of pollen grain degradation via oxidation would be worth mentioning. LINE 281: amount I have not checked the following paragraphs in detail – this should be done by a reviewer with sedimentological expertise. Concerning the interpretation itself, several parts are convincing and I appreciate how

the earlier studies are incorporated, but the aspects I discussed in the general remarks should be included. I am particularly surprised about the precise ages given in LINE 426 and LINE 439 – it is not clear to me how 1000 a BP have been determined.

Code/Data availability There are so many options to upload data in an appropriate way these days, but people can change positions, move to other countries or even change their career, therefore, it seems inappropriate to me to name one e-mail address here!

Author contribution Are all aspects mentioned here appropriate to justify being added as co-author?

Table 1: The ages at 119 and 129 have the same calibrated age (probably the one for 119 is wrong?).

Figures: It seems that genus and species names are not always in italics.

Figure 8: In addition to my problems with the age model of the core and the use of Quercus as 'temperature index', the labels in this figure are inconsistent. It should be added that the Quercus curve is from core CJ06-435 (if it is shown anyway after revision). It should be added where curve f is from. These are only a few example. . . all labels should say what it is shown and where the record is from (if the data is based on a specific core/region).

Additional references used for this review: Cheddadi, R. and Rossignol-Strick, M. 1995: Improved preservation of organic matter and pollen in Eastern Mediterranean sapropels. Paleoceanography 10, 301-309.

Havinga, A,J, 1967: Palynology and pollen preservation. Review of Palaeobotany and Palynology 2, 81-98.

Li M, Zhang S, Xu Q, Xiao J, Wen R. 2019. Spatial patterns of vegetation and climate in the North China Plain during the Last Glacial Maximum and Holocene climatic optimum. Science China Earth Sciences, 62: 1279–1287, https://doi.org/10.1007/s11430-018-9264-2

[Figure]

Apologies for some formatting probably getting lost in my comment (such as italics or superscript).

---

## Referee Comment (RC2) · Anonymous Referee #2 · 24 May 2020

This is an interesting paper that provides a better understanding about the vegetation changes during Holocene in the Bohai Sea region in response to climate change and hydrological processes. Especially it reveals that two rapid and abrupt changes in salt marsh vegetation are linked with the river-system changes. In particular, the Introduction, Geographical settings, Climate and vegetation, materials and methods and discussion are generally well written and easy to follow, but the results need to be more clear and concise and express the key findings of this study including the pollen and spore concentrations. Use ages in lieu of depths to express different pollen zones and key features as this paper is mostly focused on timescale not depth and the readers are not supposed to remember depth wise ages. Besides, considering grammar,

there are several problems with subject-verb agreement, singular and plural expressions, and less use of cohesive devices. However, these problems could be improved with an English language expert. Some specific comments are below: Page 3, line 60: As I know, the Yellow River is The Yellow River is the largest sediment transport river in the world, please check this point. Page 3, line 72: Before using Acronym for the first time is not correct. Although AMS is a very common acronym, I'll suggest to use Accelerator Mass Spectrometry (AMS) and then use AMS. Page 5, line 118: "core collection" could be substituted by "Coring". Page 5, line 123: check the acronym "NI-GLAS". Is it correct? Page 5, line 125: Did you identify the foraminifera? If so, provide their names for a better understanding. and why only 10 samples were selected? Provide an explanation. Page 6, line 132: Did you use wet or dried samples? Mention it. Page 6, line 133: Lycopodium needs to italicize. How many Lycopodium spores were in the standard tablet? Page 6, line 136: How many pollen and spore gains have you counted for each samples?

Page 6, line 134: KOH is highly corrosive and can degrade the pollen and spores if exposed for a long time. So, you need to clarify here, how long time you used the KOH. Page 6, line 137: How many pollen and spore were counted for each sample? You need to mention it. Page 6, line 138: In figure, there is CONISS. But, in this section there is no explanation of using CONISS and which software have you used for the graphs and CONISS. Make a clarification here with appropriate references. In addition, please, provide the formula used for palynomorph concentration calculation. Page 6, line 142: the expression is wrong. It should be mol/L or simply M. That is 1.0 mol/L HCl or 1.0 M HCl. Page 7, line 152: be consistent using Pb isotopic expressions throughout the manuscript. Page 7, line 170: In figures 3, 4, there are sub-zones also. Make the sentence clear by mentioning how many major and sub-zones there are. Page 7, line 172: In text it is "Palynological zone", but in Figures it is only "Zone". Be consistent using it. I'll suggest to use "Palynological zones" in the figure too. You have mentioned only the depth range. Include the ages also, like Palynological zones 1 (271–156 cm; 10000-6000 a BP). Page 8, line 176: Which type of abundance? absolute or relative? make it clear. Page 8, line 185: This sentence need to make clear. Instead of "From 156 to 128 cm..." use "From depth of 156 to 128 cm...." elsewhere. Page 9, line 224: This word "our" is less formal and overused here i this manuscript. Try to limit its use in the manuscript. There are several other expressions used instead of "our core, our study, our data, and so on". Page 11, line 269: "Figure 3 and 6e" should be replaced by "Figures 3 and 6e" as you are referencing two figures. Correct it elsewhere in the manuscript. Section 5.5 Holocene temperature variations in North China and possible driving mechanisms: Why have you chosen Quercus as a temperate index? Provide and discuss the reasons of using it as a proxy for temperate index. Section 5.4 Palaeovegetation reconstruction and its climate significance: This section need more careful considerations interpreting paleovegetation and paleoclimate. Make comparisons and combination with the findings of other nearby cores in Bohai Sea area. Although there are several cited references, they are not sufficient to establish your findings. What I mean that you need to elaborately discuss your findings and other's findings. Page 21, line 515, 517: YR in this paper has two meanings: hydrological and Yellow River. Please differentiate them.

In Table 1, the ages at depth of 119 and 129 cm are not consistent. Check and revise it. Instead of "mixed foraminifera" mention specific names and if possible the species names of them. In terms of the figures they are generally good, although you need to revise them and make more clear to understand even to a person outside of this research arena. Figure 1: Figure 1 (a) can be represented in terms of vegetation map, core locations and Figure 1 (b) can be represented along with sea bed topography to make it more interactive. Please, think about it. Figures 3 and 4: The species names are not italicized. Provide a classification of the taxa showed in the figure into trees, ferns, and herbs (upside of the graph). In addition, give a classification of arboreal, non-arboreal pollen types in the figures (may be at the bottom part). It will make the figure easier to interpret. Figure 5: There is no information about the position of land and Rivers. Point out the names in the maps for a clear understanding. Figure 8: The unit of Age is not consistent here. Sometimes you have used cal kyr BP, ka BP,

or cal. (a BP). Be consistent and use the instructions of the journal to express ages. Additional references: Ren, G. and Zhang, L., 1998. A preliminary mapped summary of Holocene pollen data for Northeast China. Quaternary Science Reviews, 17(6-7), pp.669-688. Kumar, S., Luo, C., Xiang, R., Liu, J., Chen, C. and Fang, X., 2019. High-resolution palynological record for vegetation and environment change during MIS 2 in the southern South China Sea. Marine Micropaleontology, 151(10), p.101769. Kumar, S., Luo, C., Rahman, A., Thilakanayaka, V., Khan, M.H.R., Liu, J. and Islam, G.A., 2019. Modern alluvial pollen distribution in Ganges–Brahmaputra–Meghna (GBM) floodplain and its paleoenvironmental significance. Review of Palaeobotany and Palynology, 267, pp.1-16. Li, J., Yang, S., Shu, J., Li, R., Chen, X., Meng, Y., Ye, S. and He, L., 2020. Vegetation history and environment changes since MIS 5 recorded by pollen assemblages in sediments from the western Bohai Sea, Northern China. Journal of Asian Earth Sciences, 187, p.104085.

Please also note the supplement to this comment:
https://www.clim-past-discuss.net/cp-2020-20/cp-2020-20-RC2-supplement.pdf

---

## Author Comment (AC1) · 26 Jun 2020

Major Points and reply

Question 1. Several palynological aspects are well incorporated into the manuscript, but a potential degradation of pollen grains is not appropriately discussed in my opinion. The authors correctly refer to the possible problem that there may be a transportation bias concerning bisaccate pollen (it should be considered that this may also affect Poaceae and perhaps Cyperaceae values). They also mention that the suspension of pollen grains may play an important role. But pollen concentration varies between less than 100 grains/g (which is quite low!) and several thousand grains/g in this record –

this may also point to a degradation signal. Just for example, note that Pinus pollen, which is probably much more resistant to degradation than certain nonsaccate pollen such as Quercus or Chenopodiaceae (compare e.g. Cheddai & Rossignol-Strick 1995 or Havinga 1967), increases relatively in those sections of the core which are characterized by higher sand content (and thus probably are better oxiginated). I think the aspect of pollen preservation should be mentioned in detail at the onset of the discussion and considered throughout. Answer: We thank the referee for this constructive comment. We totally agree with this comment that pollen preservation should be considered in discussion because the composition of a fossil pollen in sediment depends on several different factor including the composition of the vegetation from which the pollen originates, pollen dispersion, deposition and preservation. In a dynamic environment such as the Bohai coastal area, bias originated from bad preservation should be eliminated before using the net content of pollen grains to reconstruct paleovegetation. In this study, the pollen concentration ranged from 62 to 6050 grains/g. Relatively low pollen concentrations were found in the two sections (160–135 cm and 34–19 cm), largely correlated to high sand contents as revealed by the lithology. Especially for the lower section (150-135 cm), the high portion of sand content is consistent with a low pollen concentration and a high percentage of Pinus pollen (as the Pinus is more resistant to degradation). The variations of total pollen concentration as well as higher percentage of Pinus in this section seem to be related to pollen preservation. However, as revealed by figure 1, the highest percentage of Pinus recorded in depth of 150-128 cm, with a minor value at 145-143 cm, which is not completely in conformity with the high sand content section in the same core (160-135 cm). Similarly, for the upper section, high sand content was recorded in depth of 34-19 cm. However, the percentage of Pinus is low in this section except a relative high value in depth of 21 cm (figure 1). We thus suggest degradation is not a key point influencing the concentration of pollen and spore in the studied area. Alternatively, regional hydro-dynamic conditions may be the dominated factor for the deposition of pollen and spore. Previous research suggests that the sedimentation mechanisms of pollen and spore in marine water is similar to

that of sediment with clay- and fine silt- grainsize (Heusser, 1988). After transported to the Bohai Sea, the pollen and spore may deposit with sediment and follows the grain-size control principle. A recent investigation on the surface sediment from the Bohai Sea shows high pollen concentration in fine sediments such as clay and silty clay while low pollen concentration in coarse sediment with high percentage of sand content (figure 2; Yang et al., 2019). Yang et al. (2019) attributed the low pollen concentration in areas with a high sand content of the Bohai Sea could be attributed to the strong hydrodynamic suspension and screening for sediments and pollen. Therefore, we conclude that the low pollen concentrations in the two sections (160-135 cm, and 34-19 cm), correlated with high sand content, could be contribute to hydro-dynamic condition rather than degradation.

Figure 1: Comparison of Pinus percentage (a), sand percentage (b) and palynomorph concentration (c) in core CJ06-435 at the depth of 0-170 cm.

Figure 2: The relationship between the total pollen concentration and mud concent (%) in surface sediments of Bohai Sea, it showed that high pollen concentration were closely correlated with fine-sized sediment (Yang et al., 2019).

Heusser, L.E., 1988. Pollen distribution in marine sediments on the continental margin ofi northern California. Mar. Geol. 80, 131–147. Yang, S.X., Song, B., Ye, S.Y., Laws, E.A., He, L., Li, J., Chen, J.X., Zhao, G.M., Zhao, J.T., Mei, X., Behling, H., 2019. Large-scale pollen distribution in marine surface sediments from the Bohai Sea, China: Insights into pollen provenance, transport, deposition, and coastal-shelf paleoenvironment. Progress in Oceanography. 178, 102183.

Question 2. Another weakness is the age model, particularly in the upper part of the core. The authors dismiss three foraminifer-based ages in favor of one Cs age in the upper section of the core, which may be okay – but then it should be discussed in detail what may have caused that these ages are ca. 3000 yr too old, and how the reader can be sure that the other foraminifer-based ages are correct. It should be mentioned,

considering the problems with the uppermost ages, which foraminifer species were chosen (probably benthics?). If the used ages are correct, the 35-cm-section between 25 and 60 cm comprises more than 3000 yr, but the lower 120 cm comprise also ca. 3000 yr. This is certainly tied to the sedimentological aspects which are discussed (particularly the shift(s) of the the Yellow River Channel) by the authors, but in my opinion there remain a lot of uncertainties concerning the ages particularly above 60 cm depth. Therefore, it seems problematic to me to mention quite precise ages for the uppermost 60 cm (as done e.g. in the abstract, see below). Consider also a possible problem in Tab. 1 (see below). And how have you dealt with the older age at 119 cm compared to the younger at 129 cm – could it be redeposition? Was one age excluded? Answer: We thank the referee for this tough but constructive comment. Actually, the age model was a critical question for us, since we have tested lots of AMS14C dating points. But some of the results are confused. And, the part of age model was weak in the original MS. We rewrote the age model part in the revised manuscript and gave a detailed discussion for the current age model. Some major points are listed as follow. (1) Indeed, neither of the single species of foraminifera is enough for AMS14C dating in our core. We had to mixed all kinds of benthic foraminifera for dating (Detailed introduce was added in the revised Table 1). (2) We eliminated the dating point of 129 cm because we believed that it appears not to be reliable. According to He et al. (2019) result, the calculated sedimentation rate (CSR) in the tidal flat and neritic area of the south Bohai Sea ranged in 0.02-0.13 cm/year before ca. 2000 cal. a BP (by interpolation dating and calculation from cores H9601, H9602, ZK228, and ZK1, Figure 3). Therefore if the 129 cm dating is correct, the CSR would be as high as 0.45 cm/yr in the section of 87-129 cm (4801-4894 cal. a BP). It is apparently not reasonable because core CJ06-435 is off-shore compared to those cores (e.g. H9601, H9602, ZK228, and ZK1, Figure 3) reported in He et al. (2019) research. Thus it should have lower CSR than those cores rather than a approximate tenfold increase in CSR. (3) Based on the original age model, the record between 3000 cal. a BP and 1855 AD is somewhat confused. Because the CSR was extremely low during 3000 a BP-1855

AD (about 0.005 cm/yr). As reported in the recent study, cores from the tidal flat and neritic sea of the south Bohai Sea recorded CSR ranged in 0.02-0.13 cm/yr before 2000 cal. a BP (apparent in cores H9601, H9602, ZK228, and ZK1, Figure 3; He et al., 2019), 0.04-0.06 cm/yr between 2000 cal. a BP to 1855 AD (apparent in core ZK228, HB-1 and GYDY, Figure 3; He et al., 2019), and 0.35-1.38 cm/yr since 1855 AD (Wu et al., 2015; Qiao et al., 2017; Xu et al., 2018). Therefore, we also suggest that the primary age model covered the period of 3000 a BP-1855 AD is open to question. We guess there might be some deposition hiatus between 3000 cal. a BP and 1855 AD. Beyond that our age model and the calculated CSR in the upper unit (since 1855 AD, as calculated to 0.17-0.48 cm/yr) and the lower unit (3000-8500 cal. a BP, as calculated to 0.016-0.057 cm/yr) are comparable to the nearby records by He et al. (2019) and Xu et al. (2018). As a response, we just focused on the climate change between 3000-8500 a BP and only gave a cautious discussion for the chronology uncertain interval in the revised manuscript.

Figure 3: Locations of core CJ06-435 and other nearby cores.

He, L., Xue, C.T., Ye, S.Y., Amorosi, A., Yuan, H.M., Yang, S.X., Laws, E.A.: New evidence on the spatial-temporal distribution of superlobes in the Yellow River Delta Complex, Quat. Sci. Rev., 214, 117–138, 2019. Qiao, S.Q., Shi, X.F., Wang, G.Q., Zhou, L., Hu, B.Q., Hu, L.M., Yang, G., Liu, Y.G., Yao, Z.Q., Liu, S.F.: Sediment accumulation and budget in the Bohai Sea, Yellow Sea and East China Sea. Mar. Geol., 390, 270-281, 2017. Wu, X., Bi, N.S., Kanai, Y., Saito, Y., Zhang, Y., Yang, Z.S., Fan, D.J., Wang, H.J.: Sedimentary records off the modern Huanghe (Yellow River) delta and their response to deltaic river channel shifts over the last 200 years, J. Asian Earth Sci., 108, 68–80, 2015. Xu, Y.P., Zhou, S.Z., Hu, L.M., Wang, Y.H., Xiao, W.J.: Different controls on sedimentary organic carbon in the Bohai Sea: River mouth relocation, turbidity and eutrophication. J. Marine. Syst., 180, 1-8, 2018.

Question 3. I cannot say much about the sedimentological interpretation. Concerning the palynology-related sections of the interpretation, I think they are quite well written

(though the preservation and age model 'problems' should be considered more often, particularly when it comes to the interpretation of stages 2c to 3). I wondered in this context if the results of Li et al. 2019 would be worth being mentioned here since the record presented in the MS seems to cover the end of the Holocene climatic optimum. Answer: Thanks for this good advice and the valuable literature recommends. In the revised manuscrip Section 5.4 , we elaborately discussed our finding refering to other reported findings in north China (such as Ren and Zhang, 1998; Yi et al., 2003; Chen et al., 2012; Stebich et al., 2015; Sun and Feng, 2015; Hao et al., 2016; Li et al., 2019; Li et al., 2020; etc.) for a better interpreting of paleovegetation and paleoclimate evolution.

Question 4. What concerns me particularly when it comes to the relevance of this MS for Climate of the past is section 5.5, in which the authors link their own date to other climate records. A) Since the authors show their own data vs age (Fig. 8), it should be clearly stated how the age model was composed (linear interpolation?), it is not enough to show the dates in Tab. 1. What a about the ages at 119 and 129 cm? Answer: We are sorry for this mistake. Indeed this was an incorrect writing in Tabal.1. We have corrected this mistake in the revised manuscript. The age model was established by assuming constant sedimentation rates between two contiguous control points and extrapolation after the oldest control point.

Question 5. B) Quercus, as the authors explain, has several species in the region, therefore, the climatic susceptibility of this genus might be relatively low, and several factors, not only temperature, are influencing its relative occurrences in the pollen record. Quercus pollen is also quite susceptible to taphonomical bias (s.a.). For example, the Quercus curve of the Feng et al. (2017) record the authors cite is (naturally) completely different. C) Two other studies with pollen-based climate reconstructions which are included in Fig. 8 work with quantitative reconstructions – if the authors want their own data to be directly comparable, they should also use such an approach. The whole section 5.5 seems a little bit like an addition to make the paper a 'climate

paper". This is also consistent with some inconsistencies concerning the related Fig.8 (see below). In order to make this MS appropriate for Climate of the Past, I would suggest to use the pollen data as base for quantitative climate data. The results should be incorporated in the climate-related section. The other aspects I mentioned (taphonomy/degradation and discussion of the age model/interpolation) should be considered, too, and discussed appropriately. Answer: The authors are grateful for this constructive comment. The referee gave several useful advices which we believe can promote this study. (1) Fossil pollen spectra preserved in terrestrial sediments has been employed as a robust proxy for quantitative climate reconstructions in many regions of the world. However, quantitative palaeoclimatological estimates by marine sediments pollen data is very rare. It is probably because the quantitative reconstruction results from marine pollen data are not ideal. In the revised manuscript, we tried to use pollen data of core CJ06-435 as base for quantitative climate reconstruction. The reconstructed annual mean temperature (TANN) values ranges from 3.7oC to 6.9 oC for the past 8500 cal. a BP. However, the reconstructed TANN is much lower than the modern average annual temperature around Laizhou Bay ($\sim$12.5 oC). Therefore, we consider that the quantitative estimates results by pollen data of core CJ06-435 (a marine sediment core) may not be satisfactory. (2) Quercus has many species in the world. Different response of Quercus growth to climate in different region. Quercus mainly composed of Q. acutissima, Q. mongolica, and Q. liaotungensis in the land areas surrounding the Bohai Sea. Among these, Q. acutissima and Pinus densiflora forests develop in the low mountains and hilly area of Shandong Peninsula. Q. mongolica, Q. acutissima and P. densiflora develop in the Liaodong Peninsula (Li et al., 2007; Xu et al., 2010). It's worth noting that the pollen assemblages in marine surface sediments from the Laizhou Bay revealed that higher concentrations of Quercus and Pinus pollen distributed in the east of Laizhou Bay, and lower concentrations in the nearshore area outside the estuary of the Yellow River (Figure 4a and b). The distribution of Quercus and Pinus pollen concentration in surface sediment shows a clearly increasing shoreward the Shandong Peninsula and it is a good indicator for source tracing. In

the low mountains and hilly area of Shandong Peninsula, the vegetation is character-ized chiefly by Q. acutissima and P. densiflora forests. Modern research found that incremental temperature had positive impacts on radial growth of Q. acutissima and negative impacts on that of P. densiflora (Byun et al., 2013). For example, with the rise of annual mean temperature, Q. acutissima forests have naturally increased by approx-imately 1.13% in South Korea from 1996 to 2010, while P. densiflora decreased by 4% (Korea Forest Service, 2011; Kim et al., 2011). Therefore, the variations of Quercus and Pinus pollen from Shandong Peninsula may be related to temperature change. Except Shandong Peninsula, pollen from other regions around Laizhou Bay (such as Liaodong Peninsula) may also be transported to the Laizhou Bay, and deposited in core CJ06-435. Previous studies revealed that Quercus and Pinus were the dominant components of the forests in northeast China (including the land areas surrounding the Bohai Sea) during the Holocene. The variation of Quercus and Pinus contents were closely related to the change of temperature (Ren and Zhang, 1998; Li et al., 2004; Xu et al., 2014; Zhang et al., 2019). Ren and Zhang (1998) investigated pollen data from Northeast China and found that Quercus and Ulmus were the dominant compo-nents of the forests in northeast China between 10 and 5 ka, while Pinus were much more sparse, indicating a warmer and drier summers in northeast China for the early to mid-Holocene. A high-resolution 1000-year pollen record from the Sanjiaowan Marr Lake (42°22′16"N, 126°25′39"E) in northeastern China revealed that Quercus is a effective indicator for temperature reconstructing. Several notable cold periods, with lower Quercus frequencies, occurred at approximately 1200 AD, 1410 AD, 1580 AD, 1770 AD and 1870 AD (Zhang et al., 2019). Another 5350-year pollen record from an annually laminated maar lake (42°18.0′N, 126°21.5′E) revealed a decrease of Quercus and an increases of Pinus component, indicated a cooling trend during the past 5350 years (Xu et al., 2014). So, we suggested that Quercus is a suitable pollen type for indicating temperature variations in our study region. (3) As inferred by Referee #1 and Referee #2, Section 5.4 "Palaeovegetation reconstruction and its climate signif-icance" and Section 5.5 "Holocene temperature variations in North China and possible

driving mechanisms" need more detailed discussion. We rewrote this part in the revised manuscript. We calculated the ratio of Quercus to Pinus pollen (Q/P). Refering to previous studies such as Ren and Zhang (1998), Xu et al. (2014) and Zhang et al. (2019), the ratios of Q/P and Quercus percentage of core CJ06-435 were chosen to indicate regional temperature change, with high values indicating warm conditions. More detailed comparison of our results with global and regional climate records (we added some marine temperature records, such as Jia et al. (2019)) were presented in Section 5.5 of the revised manuscript. In Section 5.4, we gave a more detailed discussion of pollen percentage and concentration, the ecology and spread characteristics of main pollen species in core CJ06-435, paleovegetation and paleoclimate evolution of the study region and correlation and teleconnection with other findings in north China (such as Ren and Zhang, 1998; Yi et al., 2003; Chen et al., 2012; Stebich et al., 2015; Sun and Feng, 2015; Hao et al., 2016; Li et al., 2019; Li et al., 2020; etc.). We hope the revised part will be more logical and readable.

Figure 4: Spatial distribution of modern pollen percentage (black solid circle, %) and concentration (red open circle, grains/g) in Laizhou Bay, Bohai Sea (modified from Yang et al., 2016).

Byun, J.G., Lee, W.K., Kim, M., Kwak, D.A., Kwak, H., Park, T., Byun, W.H., Son, Y., Choi, J.K., Lee, Y.J., Saborowski, J., Chung, D.J., Jung, J.H.: Radial growth response of Pinus densiflora and Quercus spp. To topographic and climatic factors in South Korea, J. Plant. Ecol., 6, 380–392, 2013. Chen, W., Wang, W.M.: Middle-Late Holocene vegetation history and environment changes revealed by pollen analysis of a core at Qingdao of Shandong Province, East China, Quat. Int., 254, 68–72, 2012. Hao, Q., Liu, H.Y., Liu, X.: Pollen-detected altitudinal migration of forests during the Holocene in the mountainous forest–steppe ecotone in northern China, Palaeogeogr. Palaeoclimatol. Palaeoecol., 446, 70–77, 2016. Jia, Y.H., Li, D.W., Yu, M., Zhao, X.C., Xiang, R., Li, G.x., Zhang, H.L., Zhao, M.X.: High- and low-latitude forcing on the south Yellow Sea surface water temperature variations during the Holocene. Global. Planet.

Change., 182, 103025, 2019. Korea Forest Service: Statistical Yearbook of Forestry. Daejeon, Korea: KFS, 2011. Kim, C., Park, J.H., Jang, D.H.: Changes of the forest types by climate changes using satellite imagery and forest statistical data: a case in the Chungnam Coastal Ares, Korea, J. Environ. Impact. Asses., 20, 523–538, 2011. Li, C.Y., Yan, L.Q., Han, T.X.: Research on composition of wetland vegetation in Shandong, Shandong Forest Sci. Tech., 4, 27–29, 2007. (in Chinese with English abstract) Li, J., Yang, S.X., Shu, J.W., Li, R.H., Chen, X.H., Meng, Y.K., Ye, S.Y., He, L.: Vegetation history and environment changes since MIS 5 recorded by pollen assemblages in sediments from the western Bohai Sea, Northern China. J. Asian. Earth. Sci., 187, 104085, 2020. Li, M.Y., Zhang, S.R., Xu, Q.H., Xiao, J.L., Wen, R.L.: Spatial patterns of vegetation and climate in the North China Plain during the Last Glacial Maximum and Holocene climatic optimum. Sci. China. Earth. Sci., 62, 1279-1287, 2019. Li, X.Q., Zhou, J., Shen, J., Weng, C.Y., Zhao, H.L., Sun, Q.L.: Vegetation history and climatic variations during the last 14 ka BP inferred from a pollen record at Daihai Lake, north-central China. Rev. Palaeobot. Palyno., 132, 195– 205, 2004. Ren, G.Y., Zhang, L.S.: A preliminary mapped summary of Holocene pollen data for northeast China. Quaternary. Sci. Rev., 17, 669-688, 1998. Stebich, M., Rehfeld, K., Schlütz, F., Tarasov, P.E., Liu, J.Q., Mingram, J.: Holocene vegetation and climate dynamics of NE China based on the pollen record from Sihailongwan Maar Lake, Quat. Sci. Rev., 124, 275–289, 2015. Sun, A.Z., Feng, Z.D.: Climatic changes in the western part of the Chinese Loess Plateau during the Last Deglacial and the Holocene: A synthesis of pollen records, Quat. Int., 372, 130–141, 2015. Xu, D.K., Lu, H.Y., Chu, G.Q., Wu, N.Q., Shen, C.M., Wang, C., Mao, L.M.: 500-year climate cycles stacking of recent centennial warming documented in an East Asian pollen record. Sci. Rep., 4, 3611, 2014. Xu, Z.J., Zhang, X.L., Zhang, Z.H., Zhang, W.: Analysis of the biodiversity characters of coastal wetlands in southern Laizhou Bay, Ecol. Env. Sci., 19(2), 367–372, 2010. (in Chinese with English abstract) Yi, S., Saito, Y., Oshima, H., Zhou, Y.Q., Wei, H.L.: Holocene environmental history inferred from pollen assemblages in the Huanghe (Yellow River) delta, China: climatic change and human impact, Quat. Sci.

Rev., 22, 609–628, 2003. Zhang, J.Y., Li, J., Yan, Y., Li, J.J., Wan, X.Q.: A 1000-year record of centennial-scale cyclical vegetation change from Maar Lake Sanjiaolongwan in northeastern China. J. Asian. Earth. Sci., 176, 315–324, 2019.

Minor points: Question 6. Abstract LINES 14/15: 'Neverthesless. . . remain sparse.' This sentence implies that this is generally the case, but there are numerous studies from other regions regarding this aspect. Also 'long-term' may be confusing here since the presented record does not even span the whole Holocene. Sentences like this one might perhaps be completely removed. Answer: We agree with the advice. We have deleted this sentence in the revised manuscript.

Question 7. LINE 29 and following: If I did not completely miss anything, the age model is quite unsure between 3000 and 0 years BP (see general comments), and Pinus (excluding on peak that might be a taphonomical signal, s.a.) seems to be decreasing, compare authors' own results (4.2.3). Answer: Special thanks for this constructive advice. We agree with this comment. There may be sedimentation hiatus in the depth of 56-34 cm (3000 yr BP-1855 AD). We made a detailed answer in Question & Answer 2. In the revised manuscript, we rewrote this part and gave a new discussion.

Question 8. LINE 30: I understand this that way that the authors call the Quercus percentages a 'temperature index', which is very keen! Answer: Thanks for this good comment. We changed the sentence 'The pollen-based temperature index indicated that. . .' into 'Quercus/Pinus ratio and Quercus percentage results indicated that. . .'.

Question 9. Introduction LINES 59/60 While the abbreviations YR and BS are already explained in the abstract, maybe they should be explained again in the main text? Answer: Thanks for this good comment. We made a correction of the abbreviations YR and BS in the main text.

Question 10. Climate and vegetation LINE 103 '. . . annual mean air temperature is 7.5-14.0 oC. . .' Quite a wide range for an average temperature. Answer: Thank you. We made correction to this sentence. 'The annual mean air temperature is 9.5-13.1°C

(Qiao et al., 2012). '

Qiao, F.L., Gan, Z.J., Sun, X.P.: Regional oceanography of China seas-physical oceanography. China Ocean Press, 2012.

Question 11. LINE 109 Perhaps 'Quercus dentata'? Answer: We are sorry for the spelling error. We made correction to this mistake.

Question 12. Palynological and grain size sample analysis LINE 133: Lycopodium in italics Answer: We are sorry for the spelling error. We made correction to this mistake.

Question 13. LINE 134: Since KOH also degrades pollen, it should be mentioned how long it was used and if all samples were exposed for an identical time interval. Answer: We are sorry for the missed information. We added the detailed information in the revised manuscript. 'The samples were boiled in 10% KOH solution for 5 min to remove humic acids. '

Question 14. LINE 137: 'palynomorph sum' – is the pollen sum meant? If also dinocysts and other palynomorphs have been counted, this should be mentioned here. Answer: Thank you. 'palynomorph sum' is the pollen and spore sum meant. We have changed 'palynomorph sum' to 'sum of pollen spores'.

Question 15. LINE 137: 'exotic pollen method. . .' The whole sentence seems a little queer to me, and if Lycopod spores were used, I find the term 'pollen' misleading in this context. Answer: Thanks for this advice. We have changed this sentence to 'Pollen concentrations were calculated based on counts of Lycopodium spores added to the preparations'.

Question 16. Chronological model LINE 159: and following: It should be explained which objects were used for the dating (ideally the specific species should be mentioned). Either here or in the discussion it should be discussed what may have caused the discrepancies and why the authors trust the other AMS radiocarbon dates. Answer: Thanks for this good advice. We totally agree with the referee. We added some de-

tailed information about the materials for dating in the revision Table 1. 'The materials for dating are mixed benthic foraminifera. ' Based on the comparison of the sediment rate of our core and nearby cores, we speculate that there may be sedimentation hiatus between 3000 a BP and 1855 AD, and the age at 129 cm depth may not be reliable (Sea details in Question & Answer 2).

Question 17. Palynological Zone 1 LINE 174: 'pollen' is a singular tantum, a plural may only be appropriate if one mentions different pollen types (but even then, 'pollens' should better be avoided – occurs again later in the MS). Answer: We are sorry for the spelling error. We made correction to this mistake.

Question 18. Line 178: The MS should be consistent concerning grains/g and grains g-1. Answer: We are sorry for the spelling error. We made correction to this mistake.

Question 19. Palynological Zone 2 LINE 186: 'decline' (Plural) Answer: We are sorry for the spelling error. We made correction to this mistake.

Question 20. LINE 193: Here, NAPs may be appropriate, but I would still suggest to write NAP. Answer: We are sorry for the spelling error. We made correction to this mistake.

Question 21. LINE 197: 'percentage frequency' sounds/reads strange – percentage implies relative frequency. . . (occurs again later in the MS) Answer: We are sorry for the spelling error. We changed 'percentage frequency' to 'percentage'.

Question 22. Key terrestrial. . . LINE 237: The second sentence sounds/reads strange, and phrases like 'It is worth noting' should be avoided – if it was not worth noting, why should one mention it. Answer: Thanks . We deleted 'It is worth noting that' in the revised MS.

Question 23. LINE 238: There have been many earlier studies which revealed this effect. Perhaps it would be good to add ', also for Asian regions' or something similar after 'Previous studies', or you should cite one older study dealing with the effect. Answer: Thanks for this good advice. We cited two older study (Mudie, 1982; Mudie and Mc Carthy, 1994) to make this expression more clear.

Question 24. LINE 257: The last sentence seems useless to me. Answer: We agree with this point. We have deleted this sentence in the revised manuscript.

Question 25. Sedimentary records. . . LINE 279: I think these sentences can be significantly condensed. And in this paragraph, the aspect of pollen grain degradation via oxidation would be worth mentioning. Answer: Thanks the referee for the suggestion. We rewrote this part in the revised MS.

Question 26. LINE 281: amount I have not checked the following paragraphs in detail – this should be done by a reviewer with sedimentological expertise. Concerning the interpretation itself, several parts are convincing and I appreciate how the earlier studies are incorporated, but the aspects I discussed in the general remarks should be included. I am particularly surprised about the precise ages given in LINE 426 and LINE 439 – it is not clear to me how 1000 a BP have been determined. Answer: Thanks for this good advice. We totally agree with the referee. Please See details in Question & Answer 2.

Question 27. Code/Data availability There are so many options to upload data in an appropriate way these days, but people can change positions, move to other countries or even change their career, therefore, it seems inappropriate to me to name one e-mail address here! Answer: Thanks for this good advice. We agree with this advice. We will upload all the data of this study by the suggested gateway (such as the doi address recommended by the journal Climate of the Past) when needed.

Question 28. Author contribution are all aspects mentioned here appropriate to justify being added as co-author? Answer: Thanks for the remind. We checked the information again and confirmed all co-authors' contribution.

Question 29. Table 1: The ages at 119 and 129 have the same calibrated age (probably

the one for 119 is wrong?). Answer: We are sorry for the spelling error. We made correction to this mistake.

Question 30. Figures: It seems that genus and species names are not always in italics. Answer: We are sorry for the spelling error. We made correction to this mistake.

Question 31. Figure 8: In addition to my problems with the age model of the core and the use of Quercus as 'temperature index', the labels in this figure are inconsistent. It should be added that the Quercus curve is from core CJ06-435 (if it is shown anyway after revision). It should be added where curve f is from. These are only a few example. . . all labels should say what it is shown and where the record is from (if the data is based on a specific core/region). Answer: Thanks for this good advice. We redraw figure 8 and corrected these mistakes.

The co-authors show special thanks to referee #1 for his/her good comments and constructive advice. These comments are very valuable for improving this study.

Please also note the supplement to this comment:
https://cp.copernicus.org/preprints/cp-2020-20/cp-2020-20-AC1-supplement.pdf
* * *
[Figure]

[Figure]

**Fig. 1.**

**Fig. 2.**

[Figure]

[Figure]

**Fig. 3.**

Fig. 4.

---

## Author Comment (AC2) · 26 Jun 2020

Major Points and reply

Question 1. In particular, the Introduction, Geographical settings, Climate and vegetation, materials and methods and discussion are generally well written and easy to follow, but the results need to be more clear and concise and express the key findings of this study including the pollen and spore concentrations. Answer: Thanks for this comment. We agree with this and in the revised manuscript we rewrote some of the sections. We hope the new manuscript will be more clear and readable.

Question 2. Use ages in lieu of depths to express different pollen zones and key features as this paper is mostly focused on timescale not depth and the readers are not supposed to remember depth wise ages. Answer: Thanks for this good advice. In the revised manuscript ages information have been added into the pollen zone.

Question 3. Besides, considering grammar, there are several problems with subject-verb agreement,singular and plural expressions, and less use of cohesive devices. However, these problems could be improved with an English language expert. Answer: We are sorry for the grammar and language errors. We have invited a professional to modify the language and we have made lots of correction in the revised manuscript.

Some specific comments are below: Question 4. Page 3, line 60: As I know, the Yellow River is the largest sediment transport river in the world, please check this point. Answer: We are sorry for this ambiguous expression. According to Milliman and Meade (1983) the average sediment discharge of the Ganges Brahmaputra River, Yellow River and Amazon River are $1.67 \times 10^9$ t/a, $1.08 \times 10^9$ t/a and $0.9 \times 10^9$ t/a, respectly. While result from Meade (1996) reported that the average sediment discharge of these rivers are $(0.9\text{-}1.2) \times 10^9$ t/a (Ganges/Brahmaputra river), $1.1 \times 10^9$ t/a (Yellow River) and $(1.0\text{-}1.3) \times 10^9$ t/a (Amazon River), respectively. In the primary manuscript, we proposed that "The YR, as the second largest river in the world in terms of sediment discharge (Milliman and Meade, 1983) ". In order to avoid this inaccurate introduce, we changed the sentence "The YR, as the second largest river in the world in terms of sediment discharge" to "The Yellow River (YR), as one of the largest river in the world in terms of sediment discharge" in the revised manuscript.

Question 5. Page 3, line 72: Before using Acronym for the first time is not correct. Although AMS is a very common acronym, I'll suggest to use Accelerator Mass Spectrometry (AMS) and then use AMS. Answer: We are sorry for the spelling error. We made correction to this mistake.

Question 6. Page 5, line 118: "core collection" could be substituted by "Coring". Answer: Thanks for the Referee's careful review. We changed "core collection" to "Coring".

Question 7. Page 5, line 123: check the acronym "NIGLAS". Is it correct? Answer: We are sorry for the spelling error. The correct term is "Nanjing Institute of Geography and Limnology, Chinese Academy of Sciences (NIGLAS) ".

Question 8. Page 5, line 125: Did you identify the foraminifera? If so, provide their names for a better understanding. and why only 10 samples were selected? Provide an explanation. Answer: Actually we did no work on foraminifera identification. One importance reason is that foraminifera is quite scarce in this kind of shallow water coastal continental shelf. Indeed, neither of the single species of foraminifera is enough for AMS14C dating in our core. We had to mixed all kinds benthic foraminifera for dating (Detailed introduce was added in the revised Table 1). Even though, some of the layers are not fit for dating because there were almost no foraminifera. Therefore, we selected only 10 samples, and we have tried our best.

Question 9. Page 6, line 132: Did you use wet or dried samples? Mention it. Answer: Thanks for this good advice. All samples for pollen and spore analysis were dried at 60°C and quantified precisely. We added this message in the revised manuscript.

Question 10. Page 6, line 133: Lycopodium needs to italicize. How many Lycopodium spores were in the standard tablet? Answer: We are sorry for this mistake. The standard Lycopodium tablet (Batch 483216) is made by Lund University with a high standard. As mentioned in the Certification, each tablet contains $18,583 \pm 764$ Lycopodium spore.

Question 11. Page 6, line 136: How many pollen and spore gains have you counted for each samples? Answer: Thank you. A minimum of 200 pollen grains were counted for each sample.

Question 12. Page 6, line 134: KOH is highly corrosive and can degrade the pollen

and spores if exposed for a long time. So, you need to clarify here, how long time you used the KOH. Answer: This is a good advice. We agree the point that KOH is highly corrosive and can degrade the pollen. In order to remove humic acids in the sediment, the samples were boiled in 10% KOH solution for 5 min .We added this necessary message in the revised manuscript.

Question 13. Page 6, line 137: How many pollen and spore were counted for each sample? You need to mention it. Answer: Thank you. A minimum of 200 pollen grains were counted for each sample.

Question 14. Page 6, line 138: In figure, there is CONISS. But, in this section there is no explanation of using CONISS and which software have you used for the graphs and CONISS. Make a clarification here with appropriate references. In addition, please, provide the formula used for palynomorph concentration calculation. Answer: Thanks for this constructive comment. The pollen diagram was plotted using Tilia program. The pollen assemblage zones were divided based on the results of a constrained cluster analysis (CONISS) within Tilia. The palynomorph concentrations of per gram sediment (PCP) were calculated as the follow equation: PCP=18583/(Lycopodium number per slide)*(Pollen or Spore Counts per slide)/(Net weight of dry sample)

Question 15. Page 6, line 142: the expression is wrong. It should be mol/L or simply M. That is 1.0 mol/L HCl or 1.0 M HCl. Answer: We are sorry for the spelling error. We made correction to this mistake.

Question 16. Page 7, line 152: be consistent using Pb isotopic expressions throughout the manuscript. Answer: Thanks for this good advice. We have changed 210Pbex to excess 210Pb throughout the revised manuscript.

Question 17. Page 7, line 170: In figures 3, 4, there are sub-zones also. Make the sentence clear by mentioning how many major and sub-zones there are. Answer: Thanks for this good advice. We agree with this point. In the revised manuscript, we added the information "With the aid of CONISS, the whole sequence was vertically

divided into three zones, with zone 2 further divided into subzones 2a, 2b, 2c and 2d. "

Question18. Page 7, line 172: In text it is "Palynological zone", but in Figures it is only "Zone". Be consistent using it. I'll suggest to use "Palynological zones" in the figure too. You have mentioned only the depth range. Include the ages also, like Palynological zones 1 (271–156 cm; 10000-6000 a BP). Answer: Thanks the Referee's careful review and good advice. We used "Palynological zones" in the revised figure. Accordingly, we made some revision in the related text.

Question 19. Page 8, line 176: Which type of abundance? Absolute or relative? make it clear. Answer: We are sorry for this ambiguous expression. Here "abundance" refers to the "relative abundance". We made correction in the revised manuscript.

Question 20. Page 8, line 185: This sentence need to make clear. Instead of "From 156 to 128 cm..." use "From depth of 156 to 128 cm...." elsewhere. Answer: Thanks for this kindly advice. We changed "From ** into *** cm... " to "From depth of ** to *** cm... " in this sentence and throughout the revised manuscript.

Question 21. Page 9, line 224: This word "our" is less formal and overused here in this manuscript. Try to limit its use in the manuscript. There are several other expressions used instead of "our core, our study, our data, and so on". Answer: We appreciate the Referee for the language advice. We polished the writing of this manuscript and we hope it is more readable.

Question 22. Page 11, line 269: "Figure 3 and 6e" should be replaced by "Figures 3 and 6e" as you are referencing two figures. Correct it elsewhere in the manuscript. Answer: We are sorry for the spelling error. We made correction to this mistake.

Question 23. Section 5.5 Holocene temperature variations in North China and possible driving mechanisms: Why have you chosen Quercus as a temperate index? Provide and discuss the reasons of using it as a proxy for temperate index. Answer: We are grateful for this constructive comment. As this comment was also suggested by the

Referee #1 (Question 5, Answer (2)). Quercus has many species in the world. Different response of Quercus growth to climate in different region. Quercus mainly composed of Q. acutissima, Q. mongolica, and Q. liaotungensis in the land areas surrounding the Bohai Sea. Among these, Q. acutissima and Pinus densiflora forests develop in the low mountains and hilly area of Shandong Peninsula. Q. mongolica, Q. acutissima and P. densiflora develop in the Liaodong Peninsula (Li et al., 2007; Xu et al., 2010). It's worth noting that the pollen assemblages in marine surface sediments from the Laizhou Bay revealed that higher concentrations of Quercus and Pinus pollen distributed in the east of Laizhou Bay, and lower concentrations in the nearshore area outside the estuary of the Yellow River (Figure 1a and b). The distribution of Quercus and Pinus pollen concentration in surface sediment shows a clearly increasing shoreward the Shandong Peninsula and it is a good indicator for source tracing. In the low mountains and hilly area of Shandong Peninsula, the vegetation is characterized chiefly by Q. acutissima and P. densiflora forests. Modern research found that incremental temperature had positive impacts on radial growth of Q. acutissima and negative impacts on that of P. densiflora (Byun et al., 2013). For example, with the rise of annual mean temperature, Q. acutissima forests have naturally increased by approximately 1.13% in South Korea from 1996 to 2010, while P. densiflora decreased by 4% (Korea Forest Service, 2011; Kim et al., 2011). Therefore, the variations of Quercus and Pinus pollen from Shandong Peninsula may be related to temperature change. Except Shandong Peninsula, pollen from other regions around Laizhou Bay (such as Liaodong Peninsula) may also be transported to the Laizhou Bay, and deposited in core CJ06-435. Previous studies revealed that Quercus and Pinus were the dominant components of the forests in northeast China (including the land areas surrounding the Bohai Sea) during the Holocene. The variation of Quercus and Pinus contents were closely related to the change of temperature (Ren and Zhang, 1998; Li et al., 2004; Xu et al., 2014; Zhang et al., 2019). Ren and Zhang (1998) investigated pollen data from Northeast China and found that Quercus and Ulmus were the dominant components of the forests in northeast China between 10 and 5 ka, while Pinus were much more sparse, indicating

a warmer and drier summers in northeast China for the early to mid-Holocene. A high-resolution 1000-year pollen record from the Sanjiaowan Marr Lake (42°22′16"N, 126°25′39"E) in northeastern China revealed that Quercus is a effective indicator for temperature reconstructing. Several notable cold periods, with lower Quercus frequencies, occurred at approximately 1200 AD, 1410 AD, 1580 AD, 1770 AD and 1870 AD (Zhang et al., 2019). Another 5350-year pollen record from an annually laminated maar lake (42°18.0′N, 126°21.5′E) revealed a decrease of Quercus and an increases of Pinus component, indicated a cooling trend during the past 5350 years (Xu et al., 2014). So, we suggested that Quercus is a suitable pollen type for indicating temperature variations in our study region.

Figure 1: Spatial distribution of modern pollen percentage (black solid circle, %) and concentration (red open circle, grains/g) in Laizhou Bay, Bohai Sea (modified from Yang et al., 2016).

Question 24. Section 5.4 Palaeovegetation reconstruction and its climate significance: This section need more careful considerations interpreting paleovegetation and paleoclimate. Make comparisons and combination with the findings of other nearby cores in Bohai Sea area. Although there are several cited references, they are not sufficient to establish your findings. What I mean that you need to elaborately discuss your findings and other's findings. Answer: Thanks for this constructive comment. We agree with this point that more detailed discussion and key references should be take consideration for better interpreting paleovegetation and paleoclimate. We rewrote this part in the revised manuscript. we gave a more detailed discussion of pollen percentage and concentration, the ecology and spread characteristics of main pollen species in core CJ06-435, paleovegetation and paleoclimate evolution of the study region and correlation and teleconnection with other findings in north China (such as Ren and Zhang, 1998; Yi et al., 2003; Chen et al., 2012; Stebich et al., 2015; Sun and Feng, 2015; Hao et al., 2016; Li et al., 2019; Li et al., 2020; etc.). We hope the revised part will be more logical and readable.

Question 25. Page 21, line 515, 517: YR in this paper has two meanings: hydrological and Yellow River. Please differentiate them. Answer: Thanks for this good advice. We have changed "hydrological (YR) " in line 515 to "hydrological (the shift of YR channel) ".

Question 26. In Table 1, the ages at depth of 119 and 129 cm are not consistent. Check and revise it. Instead of "mixed foraminifera" mention specific names and if possible the species names of them. In terms of the figures they are generally good, although you need to revise them and make more clear to understand even to a person outside of this research arena. Answer: Firstly we are sorry for the mistake ("the ages at depth of 119 and 129 cm are not consistent ").We corrected this mistake in the revised manuscript. For the second question, actually we did no work on foraminifera identification. Moreover, neither of the single species of foraminifera is enough for AMS14C dating in our core. We had to mixed all kinds benthic foraminifera for dating. So, the materials for dating are mixed benthic foraminifera (Detailed introduce was added in the revised Table 1, we hope this revision is clear.).

Question 27. Figure 1: Figure 1 (a) can be represented in terms of vegetation map, core locations and Figure 1 (b) can be represented along with sea bed topography to make it more interactive. Please, think about it. Answer: Thanks for this good and constructive advice. We added an additional map illustrating the vegetation of the relevant area around the Bohai Sea in the revised MS. We redraw the figure 1(b) and added the information of water depth and topography in the revised figure 1(b).

Question 28. Figures 3 and 4: The species names are not italicized. Provide a classification of the taxa showed in the figure into trees, ferns, and herbs (upside of the graph). In addition, give a classification of arboreal, non-arboreal pollen types in the figures (may be at the bottom part). It will make the figure easier to interpret. Answer: Special thank for this kindly comment. We redraw the figures and all species names have been changed to italicized. Also, we gave classifications of arboreal, non-arboreal pollen types in the figures.

Question 29. Figure 5: There is no information about the position of land and Rivers. Point out the names in the maps for a clear understanding. Answer: Yes, we agree with the Referee. We added the information about the position of land and rivers in figure 5.

Question 30. Figure 8: The unit of Age is not consistent here. Sometimes you have used cal kyr BP, ka BP, or cal. (a BP). Be consistent and use the instructions of the journal to express ages. Answer: Special thanks for this kindly and careful review. Indeed, there were many inconsistent use of age expression throughout the primary manuscript. We are very sorry for this kind of mistakes and ignorance. In the revised manuscript, we have checked the expressions thoroughly. We hope we have eliminated all this kind of errors.

The co-authors show special thanks to Referee #2 for his/her good comments and constructive advice. These comments are very valuable for improving this study.

Please also note the supplement to this comment:
https://cp.copernicus.org/preprints/cp-2020-20/cp-2020-20-AC2-supplement.pdf

[Figure]

**Fig. 1.**